# Dimension-Free Iteration Complexity of Finite Sum Optimization Problems

**Yossi Arjevani**
Weizmann Institute of Science
Rehovot 7610001, Israel
yossi.arjevani@weizmann.ac.il

**Ohad Shamir**
Weizmann Institute of Science
Rehovot 7610001, Israel
ohad.shamir@weizmann.ac.il

## Abstract

Many canonical machine learning problems boil down to a convex optimization problem with a finite sum structure. However, whereas much progress has been made in developing faster algorithms for this setting, the inherent limitations of these problems are not satisfactorily addressed by existing lower bounds. Indeed, current bounds focus on first-order optimization algorithms, and only apply in the often unrealistic regime where the number of iterations is less than $\mathcal{O}(d/n)$ (where $d$ is the dimension and $n$ is the number of samples). In this work, we extend the framework of Arjevani et al. [3, 5] to provide new lower bounds, which are dimension-free, and go beyond the assumptions of current bounds, thereby covering standard finite sum optimization methods, e.g., SAG, SAGA, SVRG, SDCA without duality, as well as stochastic coordinate-descent methods, such as SDCA and accelerated proximal SDCA.

## 1  Introduction

Many machine learning tasks reduce to *Finite Sum Minimization (FSM)* problems of the form

$$\min_{\mathbf{w} \in \mathbb{R}^d} F(\mathbf{w}) := \frac{1}{n} \sum_{i=1}^{n} f_i(\mathbf{w}), \tag{1}$$

where $f_i$ are $L$-smooth and $\mu$-strongly convex. In recent years, a major breakthrough was made when a linear convergence rate was established for this setting (SAG [16] and SDCA [18]), and since then, many methods have been developed to achieve better convergence rate. However, whereas a large body of literature is devoted for upper bounds, the optimal convergence rate with respect to the problem parameters is not quite settled.

Let us discuss existing lower bounds for this setting, along with their shortcomings, in detail. One approach to obtain lower bounds for this setting is to consider the average of carefully handcrafted functions defined on $n$ disjoint sets of variables. This approach was taken by Agarwal and Bottou [1] who derived a lower bound for FSM under the first-order oracle model (see Nemirovsky and Yudin [12]). In this model, optimization algorithms are assumed to access a given function by issuing queries to an external first-order oracle procedure. Upon receiving a query point in the problem domain, the oracle reports the corresponding function value and gradient. The construction used by Agarwal and Bottou consisted of $n$ different quadratic functions which are adversarially determined based on the first-order queries being issued during the optimization process. The resulting bound in this case does not apply to stochastic algorithms, rendering it invalid for current state-of-the-art methods. Another instantiation of this approach was made by Lan [10] who considered $n$ disjoint copies of a quadratic function proposed by Nesterov in [13, Section 2.1.2]. This technique is based on the assumption that any iterate generated by the optimization algorithm lies in the span of previously acquired gradients.

This assumption is rather permissive and is satisfied by many first-order algorithms, e.g., SAG and SAGA [6]. However, the lower bound stated in the paper faces limitations in a few aspects. First, the validity of the derived bound is restricted to $d/n$ iterations. In many datasets, even if $d, n$ are very large, $d/n$ is quite small. Accordingly, the admissible regime of the lower bound is often not very interesting. Secondly, it is not clear how the proposed construction can be expressed as a Regularized Loss Minimization (RLM) problem with linear predictors (see Section 4). This suggests that methods specialized in dual RLM problems, such as SDCA and accelerated proximal SDCA [19], can not be addressed by this bound. Thirdly, at least the formal theorem requires assumptions (such as querying in the span of previous gradients, or sampling from a fixed distribution over the individual functions), which are not met by some state-of-the-art methods, such as coordinate descent methods, SVRG [9] and without-replacements sampling algorithms [15].

Another relevant approach in this setting is to model the functional form of the update rules. This approach was taken by Arjevani et al. [3] where new iterates are assumed to be generated by a recurrent application of some fixed linear transformation. Although this method applies to SDCA and produces a tight lower bound of $\tilde{\Omega}((n + 1/\lambda)\ln(1/\epsilon))$, its scope is rather limited. In recent work, Arjevani and Shamir [5] considerably generalized parts of this framework by introducing the class of first-order *oblivious* optimization algorithms, whose step sizes are scheduled regardless of the function under consideration, and deriving tight lower bounds for general smooth convex minimization problems (note that obliviousness rules out, e.g., quasi-Newton methods where gradients obtained at each iteration are multiplied by matrices which strictly depend on the function at hand, see Definition 2 below).

In this work, building upon the framework of oblivious algorithms, we take a somewhat more abstract point of view which allows us to easily incorporate coordinate-descent methods, as well as stochastic algorithms. Our framework subsumes the vast majority of optimization methods for machine learning problems, in particular, it applies to SDCA, accelerated proximal SDCA, SDCA without duality [17], SAG, SAGA, SVRG and acceleration schemes [7, 11]), as well as to a large number of methods for smooth convex optimization (i.e., FSM with $n = 1$), e.g., (stochastic) Gradient descent (GD), Accelerated Gradient Descent (AGD, [13]), the Heavy-Ball method (HB, [14]) and stochastic coordinate descent.

Under this structural assumption, we derive lower bounds for FSM (1), according to which the iteration complexity, i.e., the number of iterations required to obtain an $\epsilon$-optimal solution in terms of function value, is at least[1]

$$\tilde{\Omega}(n + \sqrt{n(\kappa - 1)}\ln(1/\epsilon)), \tag{2}$$

where $\kappa$ denotes the condition number of $F(\mathbf{w})$ (that is, the smoothness parameter over the strong convexity parameter). To the best of our knowledge, this is the first tight lower bound to address *all* the algorithms mentioned above. Moreover, our bound is dimension-free and thus applies to settings in machine learning which are not covered in the current literature (e.g., when $n$ is $\Omega(d)$). We also derive a dimension-free nearly-optimal lower bound for smooth convex optimization of

$$\Omega\left((L(\delta - 2)/\epsilon)^{1/\delta}\right),$$

for any $\delta \in (2, 4)$, which holds for any oblivious stochastic first-order algorithm. It should be noted that our lower bounds remain valid under any source of randomness which may be introduced into the optimization process (by the oracle or by the optimization algorithm). In particular, our bounds hold in cases where the variance of the iterates produced by the algorithm converges to zero, a highly desirable property of optimization algorithms in this setting.

Two implications can be readily derived from this lower bound. First, obliviousness forms a real barrier for optimization algorithms, and whereas non-oblivious algorithms may achieve a super-linear convergence rate at later stages of the optimization process (e.g., quasi-newton), or practically zero error after $\Theta(d)$ iterations (e.g. Center of Gravity method, MCG), oblivious algorithms are bound to linear convergence indefinitely, as demonstrated by Figure 1. We believe that this indicates that a major progress can be made in solving machine learning problems by employing non-oblivious methods for settings where $d \ll n$. It should be further noted that another major advantage of

non-oblivious algorithms is their ability to obtain optimal convergence rates without an explicit specification of the problem parameters (e.g., [5, Section 4.1]).

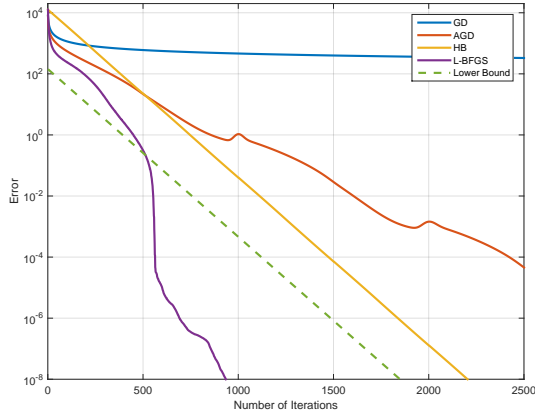

Figure 1: Comparison of first-order methods based on the function used by Nesterov in [13, Section 2.1.2] over $\mathbb{R}^{500}$. Whereas L-BFGS (with a memory size of 100) achieves a super-linear convergence rate after $\Theta(d)$ iterations, the convergence rate of GD, AGD and HB remains linear as predicted by our bound.

Secondly, many practitioners have noticed that oftentimes sampling the individual functions without replacement at each iteration performs better than sampling with replacement (e.g., [18, 15], see also [8, 20]). The fact that our lower bound holds regardless of how the individual functions are sampled and is attained using with-replacement sampling (e.g., accelerated proximal SDCA), implies that, in terms of iteration complexity, one should expect to gain no more than log factors in the problem parameters when using one method over the other (it is noteworthy that when comparing with and without replacement samplings, apart from iteration complexity, other computational resources, such as limited communication in distributed settings [4], may significantly affect the overall runtime).

## 2 Framework

### 2.1 Motivation

Due to difficulties which arise when studying the complexity of general optimization problems under discrete computational models, it is common to analyze the computational hardness of optimization algorithms by modeling the way a given algorithm interacts with the problem instances (without limiting its computational resources). In the seminal work of Nemirovsky and Yudin [12], it is shown that algorithms which access the function at hand exclusively by querying a first-order oracle require at least

$$\tilde{\Omega}\left(\min\left\{d, \sqrt{\kappa}\right\}\ln(1/\epsilon)\right), \qquad\qquad \mu > 0 \qquad\qquad (3)$$
$$\tilde{\Omega}(\min\{d\ln(1/\epsilon), \sqrt{L/\epsilon}\}), \qquad\qquad \mu = 0$$

oracle calls to obtain an $\epsilon$-optimal solution, where $L$ and $\mu$ are the smoothness and the strong convexity parameter, respectively (note that, here and throughout this section we refer to FSM problems with $n = 1$). This lower bound is tight and its dimension-free part is attained by Nesterov's well-known accelerated gradient descent, and by MCG otherwise. The fact that this approach is based on information considerations alone is very appealing and renders it valid for any first-order algorithm. However, discarding the resources needed for executing a given algorithm, in particular the per-iteration cost (in time and space), the complexity boundaries drawn by this approach are too crude from a computational point of view. Indeed, the per-iteration cost of MCG, the only method known with oracle complexity of $\mathcal{O}(d\ln(1/\epsilon))$, is excessively high, rendering it prohibitive for high-dimensional problems.

We are thus led into the question of how well can a given optimization algorithm perform assuming that its per-iteration cost is constrained? Arjevani et al. [3, 5] adopted a more structural approach

where instead of modeling how information regarding the function at hand is being collected, one models the update rules according to which iterates are being generated. Concretely, they proposed the framework of $p$-CLI optimization algorithms where, roughly speaking, new iterates are assumed to form linear combinations of the previous $p$ iterates and gradients, and the coefficients of these linear combinations are assumed to be either stationary (i.e., remain fixed throughout the optimization process) or oblivious. Based on this structural assumption, they showed that the iteration complexity of minimizing smooth and strongly convex functions is $\tilde{\Omega}(\sqrt{\kappa}\ln(1/\epsilon))$. The fact that this lower bound is stronger than (3), in the sense that it does not depend on the dimension, confirms that controlling the functional form of the update rules allows one to derive tighter lower bounds. The framework of $p$-CLIs forms the nucleus of our formulation below.

## 2.2 Definitions

When considering lower bounds one must be very precise as to the scope of optimization algorithms to which they apply. Below, we give formal definitions for oblivious stochastic CLI optimization algorithms and iteration complexity (which serves as a crude proxy for their computational complexity).

**Definition 1** (Class of Optimization Problems). *A class of optimization problems is an ordered triple $(\mathcal{F}, \mathcal{I}, \mathcal{O}_f)$, where $\mathcal{F}$ is a family of functions defined over some domain designated by $dom\mathcal{F}$, $\mathcal{I}$ is the side-information given prior to the optimization process and $\mathcal{O}_f$ is a suitable oracle which upon receiving $\mathbf{x} \in dom\mathcal{F}$ and $\theta$ in the parameter set $\Theta$, returns $\mathcal{O}_f(\mathbf{x}, \theta) \subseteq dom(\mathcal{F})$ for a given $f \in \mathcal{F}$ (we shall omit the subscript in $\mathcal{O}_f$ when $f$ is clear from the context).*

For example, in FSM, $\mathcal{F}$ contains functions as defined in (1), the side-information contains the smoothness parameter $L$, the strong convexity parameter $\mu$ and the number of components $n$ (although it carries a crucial effect on the iteration complexity, e.g., [5], in this work, we shall ignore the side-information and assume that all the parameters of the class are given). We shall assume that both first-order and coordinate-descent oracles (see 10,11 below) are allowed to be used during the optimization process. Formally, this is done by introducing an additional parameter which indicates which oracle is being addressed. This added degree of freedom does not violate our lower bounds.

We now turn to rigorously define CLI optimization algorithms. Note that, compared with the definition of first-order $p$-CLIs provided in [5], here, in order to handle coordinate-descent and first-order oracles in a unified manner, we base our formulation on general oracle procedures.

**Definition 2** (CLI). *An optimization algorithm is called a Canonical Linear Iterative (CLI) optimization algorithm over a class of optimization problems $(\mathcal{F}, \mathcal{I}, \mathcal{O}_f)$, if given an instance $f \in \mathcal{F}$ and initialization points $\{\mathbf{w}_i^{(0)}\}_{i \in \mathcal{J}} \subseteq dom(\mathcal{F})$, where $\mathcal{J}$ is some index set, it operates by iteratively generating points such that for any $i \in \mathcal{J}$,*

$$\mathbf{w}_i^{(k+1)} \in \sum_{j \in \mathcal{J}} \mathcal{O}_f\left(\mathbf{w}_j^{(k)}; \theta_{ij}^{(k)}\right), \quad k = 0, 1, \dots \tag{4}$$

*holds, where $\theta_{ij}^{(k)} \in \Theta$ are parameters chosen, stochastically or deterministically, by the algorithm, possibly depending on the side-information. If the parameters do not depend on previously acquired oracle answers, we say that the given algorithm is oblivious. Lastly, algorithms with $|\mathcal{J}| \leq p$, for some $p \in \mathbb{N}$, are denoted by $p$-CLI.*

Note that assigning different weights to different terms in (4) can be done through $\theta_{ij}^{(k)} \in \Theta$ (e.g., oracle 10 below). This allows a succinct definition for obliviousness. Lastly, we define *iteration complexity*.

**Definition 3** (Iteration Complexity). *The iteration complexity of a given CLI w.r.t. a given problem class $(\mathcal{F}, \mathcal{I}, \mathcal{O}_f)$ is defined to be the minimal number of iterations $K$ such that*

$$\mathbb{E}[f(\mathbf{w}_1^{(k)}) - \min_{\mathbf{w} \in dom\mathcal{F}} f(\mathbf{w})] < \epsilon, \quad \forall f \in \mathcal{F}, k \geq K$$

*where the expectation is taken over all the randomness introduced into the optimization process (choosing $\mathbf{w}_1^{(k)}$ merely serves as a convention and is not necessary for our bounds to hold).*

## 2.3 Proof Technique - Deriving Lower Bounds via Approximation Theory

Consider the following parametrized class of $L$-smooth and $\mu$-strongly convex optimization problems,

$$\min_{w \in \mathbb{R}} f_\eta(w) := \frac{\eta w^2}{2} - w, \quad \eta \in [\mu, L]. \tag{5}$$

Clearly, the minimizer of $f_\eta$ are $w^*(\eta) := 1/\eta$, with norm bounded by $1/\mu$. For simplicity, we will consider a special case, namely, vanilla gradient descent (GD) with step size $1/L$, which produces new iterates as follows

$$w^{(k+1)}(\eta) = w^{(k)}(\eta) - \frac{1}{L} f'_\eta(w^{(k)}(\eta)) = \left(1 - \frac{\eta}{L}\right) w^{(k)}(\eta) + \frac{1}{L}.$$

Setting the initialization point to be $w^{(0)}(\eta) = 0$, we derive an explicit expression for $w^{(k)}(\eta)$:

$$w^{(k)}(\eta) = \frac{1}{L} \sum_{i=0}^{k-1} (-1)^i \binom{k}{i+1} (\eta/L)^i. \tag{6}$$

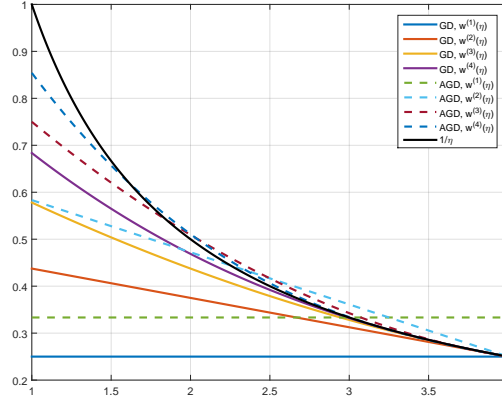

Figure 2: The first four iterates of GD and AGD, which form polynomials in $\eta$, the parameter of problem (5), are compared to $1/\eta$ over $[1, 4]$.

It turns out that each $w^{(k)}(\eta)$ forms a univariate polynomial whose degree is at most $k$. Furthermore, since $f_\eta(w)$ are $L$-smooth $\mu$-strongly convex for any $\eta \in [\mu, L]$, standard convergence analysis for GD (e.g., [13], Theorem 2.1.14) guarantees that $|w^{(k)}(\eta) - w^*(\eta)| \leq (1 - 2/(1+\kappa))^{\frac{k}{2}} |w^*(\eta)|$, where $\kappa$ denotes the condition number. Substituting Equation (6) for $w^{(k)}(\eta)$ yields

$$\max_{\eta \in [\mu, L]} \left| \frac{1}{L} \sum_{i=0}^{k-1} (-1)^i \binom{k}{i+1} (\eta/L)^i - 1/\eta \right| \leq \frac{1}{\mu} \left(1 - \frac{2}{1+\kappa}\right)^{\frac{k}{2}}.$$

Thus, we see that the faster the convergence rate of a given optimization algorithm is, the better the induced sequence of polynomials $(w^{(k)}(\eta))_{k \geq 0}$ approximate $1/\eta$ w.r.t. the maximum norm $\|\cdot\|_{L_\infty([\mu,L])}$ over $[\mu, L]$. In Fig. 2, we compare the first 4 polynomials induced by GD and AGD. Not surprisingly, AGD polynomials approximates $1/\eta$ better than those of GD.

Now, one may ask, assuming that iterates of a given optimization algorithm $\mathcal{A}$ for (5) can be expressed as polynomials $s_k(\eta)$ whose degree does not exceed the iteration number, just how fast can these iterates converge to the minimizer? Since the convergence rate is bounded from below by $\|s_k(\eta) - 1/\eta\|_{L_\infty([\mu,L])}$, we may address the following question instead:

$$\min_{s(\eta) \in \mathcal{P}_k} \|s(\eta) - 1/\eta\|_{L_\infty([\mu,L])}, \tag{7}$$

where $\mathcal{P}_k$ denotes the set of univariate polynomials whose degree does not exceed $k$. Problem (7) and other related settings are main topics of study in approximation theory. Accordingly, our technique

for proving lower bounds makes an extensive use of tools borrowed from this area. Specifically, in a paper from 1899 [21] Chebyshev showed that

$$\min_{s(\eta)\in\mathcal{P}_k}\left\|s(\eta)-\frac{1}{\eta-c}\right\|_{L_\infty([-1,1])}\geq\frac{(c-\sqrt{c^2-1})^k}{c^2-1},\quad c>1,\tag{8}$$

by which we derive the following theorem (see Appendix A.1 for a detailed proof).

**Theorem 1.** *The number of iterations required by $\mathcal{A}$ to get an $\epsilon$-optimal solution is $\tilde{\Omega}(\sqrt{\kappa}\ln(1/\epsilon))$.*

In the following sections, we apply oblivious CLI on various parameterized optimization problems so that the resulting iterates are polynomials in the problem parameters. We then apply arguments similar to the above

A similar reduction, from optimization problems to approximation problems, was used before in a few contexts to analyze the iteration complexity of deterministic CLIs (e.g., [5, Section 3], see also Conjugate Gradient convergence analysis [14]). But, what if we allow random algorithms? should we expect the same iteration complexity? To answer this, we use Yao's minimax principle according to which the performance of a given stochastic optimization algorithm w.r.t. its worst input are bounded from below by the performance of the best deterministic algorithm w.r.t. distributions over the input space. Thus, following a similar reduction one can show that the convergence rate of stochastic algorithms is bounded from below by

$$\min_{s(\eta)\in\mathcal{P}_k}\int_\mu^L|s(\eta)-1/\eta|\frac{1}{L-\mu}d\eta.\tag{9}$$

That is, a lower bound for the stochastic case can be attained by considering an approximation problem w.r.t. weighted $L_1$ with the uniform distribution over $[\mu,L]$. Other approximation problems considered in this work involve $L_2$-norm and different distributions. We provide a schematic description of our proof technique in Scheme 2.1.

| **SCHEME 2.1** | FROM OPTIMIZATION PROBLEMS TO APPROXIMATION PROBLEMS |
| --- | --- |
| **GIVEN** | A CLASS OF FUNCTIONS $\mathcal{F}$, A SUITABLE ORACLE $\mathcal{O}$ |
| | AND A SEQUENCE OF SETS OF FUNCTION $\mathcal{S}_k$ OVER SOME PARAMETERS SET $H$. |
| **CHOOSE** | A SUBSET OF FUNCTIONS $\{f_\eta\in\mathcal{F}\|\eta\in H\}$, S.T. $\mathbf{w}^k(\eta)\in\mathcal{S}_k$. |
| **COMPUTE** | THE MINIMIZER $\mathbf{w}^*(\eta)$ FOR ANY $f_\eta$ |
| **BOUND** | FROM BELOW THE BEST APPROXIMATION FOR $\mathbf{w}^*(\eta)$ W.R.T. $\mathcal{S}_k$ |
| | AND A NORM $\|\cdot\|$, I.E., $\min\{\|\mathbf{s}(\eta)-\mathbf{w}^*(\eta)\|\mid\mathbf{s}(\eta)\in\mathcal{S}_k\}$ |

## 3 Lower Bound for Finite Sums Minimization Methods

Having described our analytic approach, we now turn to present some concrete applications, starting with iteration complexity lower bounds in the context of FSM problems (1). In what follows, we derive a lower bound on the iteration complexity of oblivious (possibly stochastic) CLI algorithms equipped with first-order and coordinate-descent oracles for FSM. Strictly speaking, we focus on optimization algorithms equipped with both generalized first order oracle,

$$\mathcal{O}(\mathbf{w};A,B,\mathbf{c},j)=A\nabla f_j(\mathbf{w})+B\mathbf{w}+\mathbf{c},\quad A,B\in\mathbb{R}^{d\times d},\mathbf{c}\in\mathbb{R}^d,j\in[n],\tag{10}$$

and steepest coordinate-descent oracle

$$\mathcal{O}(\mathbf{w};i,j)=\mathbf{w}+t^*\mathbf{e}_i,\quad t^*\in\underset{t\in\mathbb{R}}{\operatorname{argmin}}f_j(w_1,\ldots,w_{i-1},w_i+t,w_{i+1},\ldots,w_d),j\in[n],\tag{11}$$

where $\mathbf{e}_i$ denotes the $i$'th unit vector. We remark that coordinate-descent steps w.r.t. partial gradients can be implemented using (10) by setting $A$ to be some principal minor of the unit matrix. It should be further noted that our results below hold for scenarios where the optimization algorithm is free to call a different oracle at different iterations.

First, we sketch the proof of the lower bound for deterministic oblivious CLIs. Following Scheme 2.1, we restrict our attention to a parameterized subset of problems. We assume[2] $d>1$ and denote by

$\mathcal{H}_{\text{FSM}}$ the set of all $(\eta_1, \ldots, \eta_n) \in \mathbb{R}^n$ such that all the entries equal $-(L-\mu)/2$, except for some $j \in [n]$, for which $\eta_j \in [-(L-\mu)/2, (L-\mu)/2]$. Now, given $\boldsymbol{\eta} := (\eta_1, \ldots, \eta_n) \in \mathcal{H}_{\text{FSM}}$ we define

$$F_{\boldsymbol{\eta}}(\mathbf{w}) := \frac{1}{n} \sum_{i=1}^{n} \left( \frac{1}{2} \mathbf{w}^\top Q_{\eta_i} \mathbf{w} - \mathbf{q}^\top \mathbf{w} \right), \text{where} \tag{12}$$

$$Q_{\eta_i} := \begin{pmatrix} \frac{L+\mu}{2} & \eta_i & & & \\ \eta_i & \frac{L+\mu}{2} & & & \\ & & \mu & & \\ & & & \ddots & \\ & & & & \mu \end{pmatrix}, \quad \mathbf{q} := \begin{pmatrix} \frac{R\mu}{\sqrt{2}} \\ \frac{R\mu}{\sqrt{2}} \\ 0 \\ \vdots \\ 0 \end{pmatrix}.$$

It is easy to verify that the minimizers of (12) are

$$\mathbf{w}^*(\boldsymbol{\eta}) = \left( \frac{R\mu}{\sqrt{2} \left( \frac{L+\mu}{2} + \frac{1}{n} \sum_{i=1}^{n} \eta_i \right)}, \frac{R\mu}{\sqrt{2} \left( \frac{L+\mu}{2} + \frac{1}{n} \sum_{i=1}^{n} \eta_i \right)}, 0, \ldots, 0 \right)^\top. \tag{13}$$

We would like to show that the coordinates of the iterates of deterministic oblivious CLIs, which minimize $F_{\boldsymbol{\eta}}$ using first-order and coordinate-descent oracles, form multivariate polynomials in $\boldsymbol{\eta}$ of total degrees (the maximal sum of powers over all the terms) which does not exceed the iteration number. Indeed, if the coordinates of $\mathbf{w}_i^{(k)}(\boldsymbol{\eta})$ are multivariate polynomial in $\boldsymbol{\eta}$ of total degree at most $k$, then the coordinates of the vectors returned by both oracles

First-order oracle: $\mathcal{O}(\mathbf{w}_j^{(k)}; A, B, \mathbf{c}, j) = A(Q_{\eta_j} \mathbf{w}_i^{(k)} - \mathbf{q}) + B\mathbf{w}_i^{(k)} + \mathbf{c}$, $\tag{14}$

Coordinate-descent oracle: $\mathcal{O}(\mathbf{w}_j^{(k)}; i, j) = \left( I - (1/(Q_{\eta_j})_{ii}) \mathbf{e}_i (Q_{\eta_j})_{i,*} \right) \mathbf{w}_i^{(k)} - q_i/(Q_{\eta_j})_{ii} \mathbf{e}_i$,

are multivariate polynomials of total degree of at most $k+1$, as all the parameters ($A, B, C, i$ and $j$) do not depend on $\boldsymbol{\eta}$ (due to obliviousness) and the rest of the terms ($Q_{\eta_j}, \mathbf{q}, I, 1/(Q_{\eta_j})_{ii}, (Q_{\eta_j})_{i,*}, \mathbf{e}_i$ and $q_i$) are either linear in $\eta_j$ or constants. Now, since the next iterates are generated simply by summing up all the oracle answers, they also form multivariate polynomials of total degree of at most $k+1$. Thus, denoting the first coordinate of $\mathbf{w}_1^{(k)}(\boldsymbol{\eta})$ by $s(\boldsymbol{\eta})$ and using Inequality (8), we get the following bound

$$\max_{\boldsymbol{\eta} \in \mathcal{H}_{\text{FSM}}} \|\mathbf{w}_1^{(k)}(\boldsymbol{\eta}) - \mathbf{w}^*(\boldsymbol{\eta})\| \geq \left\| s(\boldsymbol{\eta}) - \frac{R\mu}{\sqrt{2} \left( \frac{L+\mu}{2} + \frac{1}{n} \sum_{i=1}^{n} \eta_i \right)} \right\|_{L^\infty([\mu,L])} \tag{15}$$

$$\geq \Omega(1) \left( \frac{\sqrt{\frac{\kappa-1}{n}+1} - 1}{\sqrt{\frac{\kappa-1}{n}+1} + 1} \right)^{k/n}, \tag{16}$$

where $\Omega(1)$ designates a constant which does not depend on $k$ (but may depend on the problem parameters). Lastly, this implies that for any deterministic oblivious CLI and any iteration number, there exists some $\boldsymbol{\eta} \in \mathcal{H}_{\text{FSM}}$ such that the convergence rate of the algorithm, when applied on $F_{\boldsymbol{\eta}}$, is bounded from below by Inequality (16). We note that, as opposed to other related lower bounds, e.g., [10], our proof is non-constructive. As discussed in subsection 2.3, this type of analysis can be extended to stochastic algorithms by considering (15) w.r.t. other norms such as weighted $L_1$-norm. We now arrive at the following theorem whose proof, including the corresponding logarithmic factors and constants, can be found in Appendix A.2.

**Theorem 2.** *The iteration complexity of oblivious (possibly stochastic) CLIs for FSM (1) equipped with first-order (10) and coordinate-descent oracles (11), is bounded from below by*

$$\tilde{\Omega}(n + \sqrt{n(\kappa-1)} \ln(1/\epsilon)).$$

The lower bound stated in Theorem 2 is tight and is attained by, e.g., SAG combined with an acceleration scheme (e.g., [11]). Moreover, as mentioned earlier, our lower bound does not depend on the problem dimension (or equivalently, holds for any number of iterations, regardless of $d$ and

$n$), and covers coordinate descent methods with stochastic or deterministic coordinate schedule (in the special case where $n = 1$, this gives a lower bound for minimizing smooth and strongly convex functions by performing steepest coordinate descent steps). Also, our bound implies that using mini-batches for tackling FSM does not reduce the overall iteration complexity. Lastly, it is noteworthy that the $n$ term in the lower bound above holds for any algorithm accompanied with an *incremental oracle*, which grants access to at most one individual function each time.

We also derive a nearly-optimal lower bound for smooth non-strongly convex functions for the more restricted setting of $n = 1$ and first-order oracle. The parameterized subset of functions we use (see Scheme 2.1) is $g_\eta(\mathbf{x}) := \frac{\eta}{2} \|\mathbf{x}\|^2 - R\eta \mathbf{e}_1^\top \mathbf{x}, \quad \eta \in (0, L]$. The corresponding minimizer (as a function of $\eta$) is $\mathbf{x}^*(\eta) = R\mathbf{e}_1$, and in this case we seek to approximate it w.r.t. $L_2$-norm using $k$-degree univariate polynomials whose constant term vanishes. The resulting bound is dimension-free and improves upon other bounds for this setting (e.g. [5]) in that it applies to deterministic algorithms, as well as to stochastic algorithms (see A.3 for proof).

**Theorem 3.** *The iteration complexity of any oblivious (possibly stochastic) CLI for $L$-smooth convex functions equipped with a first-order oracle, is bounded from below by*

$$\Omega\left((L(\delta - 2)/\epsilon)^{1/\delta}\right), \ \delta \in (2, 4).$$

## 4 Lower Bound for Dual Regularized Loss Minimization with Linear Predictors

The form of functions (12) discussed in the previous section does not readily adapt to general RLM problems with linear predictors, i.e.,

$$\min_{\mathbf{w} \in \mathbb{R}^d} P(\mathbf{w}) := \frac{1}{n} \sum_{i=1}^{n} \phi_i(\langle \mathbf{x}_i, \mathbf{w} \rangle) + \frac{\lambda}{2} \|\mathbf{w}\|^2, \tag{17}$$

where the loss functions $\phi_i$ are $L$-smooth and convex, the samples $\mathbf{x}_1, \ldots, \mathbf{x}_n$ are $d$-dimensional vectors in $\mathbb{R}^d$ and $\lambda$ is some positive constant. Thus, dual methods which exploit the added structure of this setting through the *dual problem* [18],

$$\min_{\boldsymbol{\alpha} \in \mathbb{R}^n} D(\boldsymbol{\alpha}) = \frac{1}{n} \sum_{i=1}^{n} \phi_i^*(-\alpha_i) + \frac{\lambda}{2} \left\| \frac{1}{\lambda n} \sum_{i=1}^{n} \mathbf{x}_i \alpha_i \right\|^2, \tag{18}$$

such as SDCA and accelerated proximal SDCA, are not covered by Theorem 2. Accordingly, in this section, we address the iteration complexity of oblivious (possibly stochastic) CLI algorithms equipped with dual RLM oracles:

$$\mathcal{O}(\boldsymbol{\alpha}; t, j) = \boldsymbol{\alpha} + t\nabla_j D(\boldsymbol{\alpha})\mathbf{e}_j, \quad t \in \mathbb{R}, j \in [n], \tag{19}$$
$$\mathcal{O}(\boldsymbol{\alpha}; j) = \boldsymbol{\alpha} + t^*\mathbf{e}_j, \quad t^* = \underset{t \in \mathbb{R}}{\operatorname{argmin}} D(\alpha_1, \ldots, \alpha_{j-1}, \alpha_j + t, \alpha_{j+1}, \ldots, \alpha_d), j \in [n],$$

Following Scheme 2.1, we first describe the relevant parametrized subset of RLM problems. For the sake of simplicity, we assume that $n$ is even (the proof for odd $n$ holds mutandis mutatis). We denote by $\mathcal{H}_{\text{RLM}}$ the set of all $(\psi_1, \ldots, \psi_{n/2}) \in \mathbb{R}^{n/2}$ such that all entries are 0, except for some $j \in [n/2]$, for which $\psi_j \in [-\pi/2, \pi/2]$. Now, given $\boldsymbol{\psi} \in \mathcal{H}_{\text{RLM}}$, we set $P_{\boldsymbol{\psi}}$ (defined in 17) as follows

$$\phi_i(w) = \frac{1}{2}(w+1)^2, \quad \mathbf{x}_{\boldsymbol{\psi},i} = \begin{cases} \cos(\psi_{(i+1)/2})\mathbf{e}_i + \sin(\psi_{(i+1)/2})\mathbf{e}_{i+1} & i \text{ is odd} \\ \mathbf{e}_i & \text{o.w.} \end{cases}.$$

We state below the corresponding lower bound, whose proof, including logarithmic factors and constants, can be found in Appendix A.4.

**Theorem 4.** *The iteration complexity of oblivious (possibly stochastic) CLIs for RLM (17) equipped with dual RLM oracles (19) is bounded from below by*

$$\tilde{\Omega}(n + \sqrt{nL/\lambda} \ln(1/\epsilon)).$$

This bound is tight w.r.t. the class of oblivious CLIs and is attained by accelerated proximal SDCA. As mentioned earlier, a tighter lower bound of $\tilde{\Omega}((n + 1/\lambda) \ln(1/\epsilon))$ is known for SDCA [3], suggesting that a tighter bound might hold for the more restricted set of stationary CLIs (for which the oracle parameters remain fixed throughout the optimization process).

## Footnotes

[1]Following standard conventions, here tilde notation hides logarithmic factors in the parameters of a given class of optimization problems, e.g., smoothness parameter and number of components.

[2]Clearly, in order to derive a lower bound for coordinate-descent algorithms, we must assume $d>1$. If only a first-order oracle is allowed, then the same lower bound as in Theorem 2 can be derived for $d=1$.
