[Supplementary Material]

# A   Proofs

## A.1   Proof of Theorem 1

**Proof** According to the way $\mathcal{A}$ generates iterates, we have

$$|w^{(k)}(\eta) - w^*(\eta)| = |s_k(\eta) - 1/\eta|, \quad \eta \in [\mu, L]$$

for some polynomial $s_k(\eta)$ of degree at most $k$. By Lemma 6, we have

$$\min_{s(\eta) \in \mathcal{P}_k} \left\| s(\eta) - \frac{1}{\eta} \right\|_{L_\infty([\mu, L])} \geq \frac{L - \mu}{2L\mu} \left( \frac{\sqrt{\kappa} - 1}{\sqrt{\kappa} + 1} \right)^k,$$

where $\kappa = L/\mu$. Thus,

$$|w^{(k)}(\eta) - w^*(\eta)| \geq \min_{s(\eta) \in \mathcal{P}_k} \left\| s(\eta) - \frac{1}{\eta} \right\|_{L_\infty([\mu, L])} \geq \frac{L - \mu}{2L\mu} \left( \frac{\sqrt{\kappa} - 1}{\sqrt{\kappa} + 1} \right)^k \geq |w^*(\eta)| \frac{L - \mu}{2L} \left( \frac{\sqrt{\kappa} - 1}{\sqrt{\kappa} + 1} \right)^k.$$

Now, since $f_\eta$ is $\mu$-strongly convex, we have,

$$
\begin{aligned}
f(w^{(k)}(\eta)) - f(w^*(\eta)))| &\geq \frac{\mu}{2} |w^{(k)}(\eta) - w^*(\eta)|^2 \\
&\geq \frac{\mu}{2} \left( |w^*(\eta)| \frac{L - \mu}{2L} \left( \frac{\sqrt{\kappa} - 1}{\sqrt{\kappa} + 1} \right)^k \right)^2 \\
&= \frac{\mu}{2} \left( |w^*(\eta)| \frac{L - \mu}{2L} \right)^2 \left( \frac{\sqrt{\kappa} - 1}{\sqrt{\kappa} + 1} \right)^{2k}.
\end{aligned}
$$

Hence, by Lemma 12, the minimal number of iterations required to get an $\epsilon$-optimal solution is at least

$$\frac{1}{4} \sqrt{\kappa - 1} \left( \ln \frac{\mu}{2} + 2 \ln \left( |w^*(\eta)| \frac{L - \mu}{2L} \right) + \ln(1/\epsilon) \right).$$

∎

## A.2   Proof of Theorem 2 - Finite Sums

When dealing with multivariate polynomials it is convenient to define multi-indices $\mathbf{i} = (i_1, \ldots, i_n) \in \mathbb{N}_0^n$, where $\mathbb{N}_0^n$ is the set of all $n$-tuples of non-negative integers. In addition, with a slight abuse of notation, we define

$$\mathcal{P}_k^n := \mathrm{span} \left\{ \eta^{\mathbf{i}} \mid \mathbf{i} \in \mathbb{N}_0^n, \ |\mathbf{i}| \leq k \right\}, \tag{20}$$

where we put $\eta^{\mathbf{i}} = \eta_1^{i_1} \cdots \eta_n^{i_n}$ and $|\mathbf{i}| = i_1 + \cdots + i_n$. In words, $\mathcal{P}_k^n$ is the set of all multivariate polynomials over $n$ indeterminates whose total degree (the maximal sum of the degrees over all terms) is less than or equal to $k$. Lastly, given $\mathbf{s}(\boldsymbol{\eta}) \in \mathcal{P}_k^n$ we define

$$s_i(\eta_i) := \mathbf{s} \left( -\frac{L - \mu}{2}, \ldots, -\frac{L - \mu}{2}, \underbrace{\eta_i}_{i\text{'th entry}}, -\frac{L - \mu}{2}, \ldots, -\frac{L - \mu}{2} \right).$$

This notation will come in handy in the main proof.

The lemma below describes the functional form assumed by iterates produced by oblivious CLIs.

**Lemma 1.** *When applied on (12) with suitable first-order and coordinate-descent oracles (as defined in 14), the coordinates of iterates produced by oblivious stochastic CLIs form multivariate polynomials in $\boldsymbol{\eta}$ with random real coefficients whose total degree does not exceed the iteration number.*

**Proof** Let $\mathcal{A}$ be an oblivious stochastic CLI, and suppose we apply $\mathcal{A}$ on the class of problems (12) parameterized by $\boldsymbol{\eta}$, using both first-order and coordinate-descent oracles as defined in 14. We use mathematical induction to show that for any $k = 0, 1, \ldots$, the coordinate of the $k$'th iterate produced by such process can be expressed as a distribution over multivariate polynomials in $\boldsymbol{\eta}$ of degree at most $k$.

As the first iterate $\mathbf{w}_i^{(0)}$ is allowed to depend only on $L, \mu$ and $n$, the base case is trivial. That is, the coordinates of $\mathbf{w}_i^{(0)}$ form distributions over $\mathbb{R} = \mathcal{P}_0^n$ which do not depend on $\boldsymbol{\eta}$.

For the inductive step, assume that any coordinate of $\mathbf{w}_i^{(k)}(\boldsymbol{\eta})$ can be expressed as a distribution over $\mathcal{P}_k^n$. It is easy to see that for any $\mathbf{w}_i^{(k)}(\boldsymbol{\eta})$, the answers of both oracles,

First-order oracle: $\qquad \mathcal{O}(\mathbf{w}_i^{(k)}; A, B, \mathbf{c}, j) = A(Q_{\eta_j} \mathbf{w}_i^{(k)} - \mathbf{q}) + B\mathbf{w}_i^{(k)} + \mathbf{c},$

Coordinate-descent oracle: $\quad \mathcal{O}(\mathbf{w}_i^{(k)}; i, j) = \left(I - (1/(Q_{\eta_j})_{ii})\mathbf{e}_i(Q_{\eta_j})_{i,*}\right)\mathbf{w}_i^{(k)} - q_i/(Q_{\eta_j})_{ii}\mathbf{e}_i,$

form a distribution over $\mathcal{P}_{k+1}^n$, as all the random quantities involved in the expressions $(A, B, \mathbf{c}, i$ and $j)$ do not depend on $\eta_1, \ldots, \eta_n$ (due to obliviousness) and the rest of the terms $(I, Q_{\eta_j}, 1/(Q_{\eta_j})_{ii}, (Q_{\eta_j})_{i,*}, \mathbf{e}_i, q_i$ and $\mathbf{q})$ are either linear in $\eta_j$ or constants. Lastly, $\mathbf{w}_i^{(k+1)}$ are computed by simply summing up all the oracle answers, and as such, form again distributions over $\mathcal{P}_{k+1}^n$. ■

**Proof** [Theorem 2] Let $\mathcal{A}$ be an oblivious stochastic CLI. By Lemma 1 the first coordinate of $\mathbf{w}_1^{(k)}(\boldsymbol{\eta})$ (the point returned by the algorithm at the $k$'th iteration) when applied on class of problems (12) distributes according to some distribution $\mathcal{D}$ over $\mathcal{P}_k^n$. Thus, by Yao principle, since each polynomial in $(\mathcal{P}_k^n)^d$ represents a single deterministic algorithm, we have

$$\max_{\boldsymbol{\eta} \in \mathcal{H}} \mathbb{E}_{\mathbf{w}_1^{(k)}(\boldsymbol{\eta}) \sim \mathcal{D}} \|\mathbf{w}_1^{(k)}(\boldsymbol{\eta}) - \mathbf{w}^*(\boldsymbol{\eta})\| \geq \min_{\mathbf{s}(\boldsymbol{\eta}) \in (\mathcal{P}_k^n)^d} \mathbb{E}_{\boldsymbol{\eta} \sim \mathcal{U}(\mathcal{H})} \|\mathbf{s}(\boldsymbol{\eta}) - \mathbf{w}^*(\boldsymbol{\eta})\| \qquad (21)$$

where $\mathcal{U}(\mathcal{H})$ denotes a distribution over $\mathcal{H}$ which corresponds to first drawing $j \sim \mathcal{U}([n])$ at random, and then setting the coordinates of $\boldsymbol{\eta}$ as follows

$$\begin{cases} \eta_i \sim \mathcal{U}([-(L-\mu)/2, (L-\mu)/2] & i = j \\ \eta_i = -\frac{L-\mu}{2}, & i \neq j \end{cases}. \qquad (22)$$

Furthermore, it is easy to verify that the corresponding minimizers of (12) are

$$\mathbf{w}^*(\eta_1, \ldots, \eta_n) = \left(\frac{1}{n} \sum_{i=1}^n Q_{\eta_i}\right)^{-1} \mathbf{q} = \left(\frac{R\mu}{\sqrt{2}\left(\frac{L+\mu}{2} + \frac{1}{n}\sum_{i=1}^n \eta_i\right)}, \frac{R\mu}{\sqrt{2}\left(\frac{L+\mu}{2} + \frac{1}{n}\sum_{i=1}^n \eta_i\right)}, 0, \ldots, 0\right)^\top. \qquad (23)$$

We now have,

$$\min_{\mathbf{s}(\boldsymbol{\eta}) \in (\mathcal{P}_k^n)^d} \mathbb{E}_{\boldsymbol{\eta} \sim \mathcal{U}(\mathcal{H})} \|\mathbf{s}(\boldsymbol{\eta}) - \mathbf{w}^*(\boldsymbol{\eta})\| = \min_{\mathbf{s}(\boldsymbol{\eta}) \in (\mathcal{P}_k^n)^d} \mathbb{E}_{i \sim \mathcal{U}([n])} \mathbb{E}_{\eta_i \sim \mathcal{U}([-\frac{L-\mu}{2}, \frac{L-\mu}{2}])} \|\mathbf{s}(\boldsymbol{\eta}) - \mathbf{w}^*(\boldsymbol{\eta})\|$$

$$\geq \frac{1}{n} \min_{s(\boldsymbol{\eta}) \in \mathcal{P}_k^n} \sum_{j=1}^n \mathbb{E}_{\eta_j \sim \mathcal{U}([-\frac{L-\mu}{2}, \frac{L-\mu}{2}])} \left| s_j(\eta_j) - \frac{R\mu}{\sqrt{2}(\frac{1}{n}\sum_{i=1}^n \eta_i + \frac{L+\mu}{2})} \right|$$

$$= \frac{R\mu}{\sqrt{2}} \min_{s(\boldsymbol{\eta}) \in \mathcal{P}_k^n} \sum_{j=1}^n \mathbb{E}_{\eta_j \sim \mathcal{U}([-\frac{L-\mu}{2}, \frac{L-\mu}{2}])} \left| s_j(\eta_j) - \frac{1}{\eta_j - (n-1)\frac{L-\mu}{2} + n\frac{L+\mu}{2}} \right|$$

$$= \frac{R\mu}{\sqrt{2}} \min_{s(\boldsymbol{\eta}) \in \mathcal{P}_k^n} \sum_{j=1}^n \int_{-\frac{L-\mu}{2}}^{\frac{L-\mu}{2}} \left| s_j(\eta_j) - \frac{1}{\eta_j - (n-1)\frac{L-\mu}{2} + n\frac{L+\mu}{2}} \right| \frac{1}{L-\mu} d\eta_j$$

$$= \frac{R\mu}{\sqrt{2}(L-\mu)} \min_{s(\boldsymbol{\eta}) \in \mathcal{P}_k^n} \sum_{j=1}^n \int_{-\frac{L-\mu}{2}}^{\frac{L-\mu}{2}} \left| s_j(\eta_j) - \frac{1}{\eta_j - (n-1)\frac{L-\mu}{2} + n\frac{L+\mu}{2}} \right| d\eta_j \qquad (24)$$

where the first inequality follows by focusing on the first coordinate of $\mathbf{s}(\boldsymbol{\eta}) - \mathbf{w}^*(\boldsymbol{\eta})$. Now, set $\alpha = -(n-1)\frac{L-\mu}{2} + n\frac{L+\mu}{2}$ and note that

$$\sqrt{\frac{2\alpha + L - \mu}{2\alpha + \mu - L}} = \sqrt{\frac{2(-(n-1)\frac{L-\mu}{2} + n\frac{L+\mu}{2}) + L - \mu}{2(-(n-1)\frac{L-\mu}{2} + n\frac{L+\mu}{2}) + \mu - L}} = \sqrt{\frac{\kappa-1}{n} + 1}.$$

Thus, by Lemma 8 (using the same value for $\alpha$ and noting that $\alpha > (L-\mu)/2$) yields

$$\int_{-\frac{L-\mu}{2}}^{\frac{L-\mu}{2}} \left| s_j(\eta_j) - \frac{1}{\eta_j - (n-1)\frac{L-\mu}{2} + n\frac{L+\mu}{2}} \right| d\eta_j \geq \left( \frac{\sqrt{\frac{\kappa-1}{n} + 1} - 1}{\sqrt{\frac{\kappa-1}{n} + 1} + 1} \right)^{k_j}.$$

where $k_j$ denotes the degree of $s_j(\eta_j)$. Plugging in this into Inequality (24) we get

$$\max_{\boldsymbol{\eta} \in \mathcal{H}} \mathbb{E}_{\mathbf{w}_1^{(k)}(\boldsymbol{\eta}) \sim \mathcal{D}} \|\mathbf{w}_1^{(k)}(\boldsymbol{\eta}) - \mathbf{w}^*(\boldsymbol{\eta})\| \geq \frac{nR\mu}{\sqrt{2}(L-\mu)} \min_{\mathbf{s}(\boldsymbol{\eta}) \in \mathcal{P}_k^n} \frac{1}{n} \sum_{j=1}^{n} \left( \frac{\sqrt{\frac{\kappa-1}{n} + 1} - 1}{\sqrt{\frac{\kappa-1}{n} + 1} + 1} \right)^{k_j}.$$

Since $u \mapsto \rho^u$ is a decreasing and convex function for any $1 > \rho > 0$, we have

$$\frac{nR\mu}{\sqrt{2}(L-\mu)} \min_{\mathbf{s}(\boldsymbol{\eta}) \in \mathcal{P}_k^n} \frac{1}{n} \sum_{j=1}^{n} \left( \frac{\sqrt{\frac{\kappa-1}{n} + 1} - 1}{\sqrt{\frac{\kappa-1}{n} + 1} + 1} \right)^{k_j} \geq \frac{nR\mu}{\sqrt{2}(L-\mu)} \min_{\mathbf{s}(\boldsymbol{\eta}) \in \mathcal{P}_k^n} \left( \frac{\sqrt{\frac{\kappa-1}{n} + 1} - 1}{\sqrt{\frac{\kappa-1}{n} + 1} + 1} \right)^{\frac{1}{n}\sum_{j=1}^{n} k_j}$$

$$\geq \frac{nR\mu}{\sqrt{2}(L-\mu)} \left( \frac{\sqrt{\frac{\kappa-1}{n} + 1} - 1}{\sqrt{\frac{\kappa-1}{n} + 1} + 1} \right)^{k/n}$$

where the last inequality is due to the fact that $\mathbf{s}(\boldsymbol{\eta}) \in \mathcal{P}_k^n$ which implies that $\sum_{j=1}^{n} k_j \leq k$. Finally, we have,

$$\max_{\boldsymbol{\eta} \in \mathcal{H}} \mathbb{E}_{\mathbf{w}_1^{(k)}(\boldsymbol{\eta}) \sim \mathcal{D}} [F_{\boldsymbol{\eta}}(\mathbf{w}_1^{(k)}(\boldsymbol{\eta})) - F_{\boldsymbol{\eta}}(\mathbf{w}^*(\boldsymbol{\eta}))] \geq \max_{\boldsymbol{\eta} \in \mathcal{H}} \mathbb{E}_{\mathbf{w}_1^{(k)}(\boldsymbol{\eta}) \sim \mathcal{D}} \frac{\mu}{2} \|\mathbf{w}_1^{(k)}(\boldsymbol{\eta}) - \mathbf{w}^*(\boldsymbol{\eta})\|^2$$

$$\geq \frac{\mu}{2} \left( \frac{nR\mu}{\sqrt{2}(L-\mu)} \left( \frac{\sqrt{\frac{\kappa-1}{n} + 1} - 1}{\sqrt{\frac{\kappa-1}{n} + 1} + 1} \right)^{k/n} \right)^2$$

$$= \frac{\mu}{2} \left( \frac{nR\mu}{\sqrt{2}(L-\mu)} \right)^2 \left( \frac{\sqrt{\frac{\kappa-1}{n} + 1} - 1}{\sqrt{\frac{\kappa-1}{n} + 1} + 1} \right)^{2k/n}$$

where the first inequality follows by the $\mu$-strong convexity of $F_{\boldsymbol{\eta}}$ and the second inequality follows by Jensen inequality. Using Lemma 12, we get that the iteration complexity of $\mathcal{A}$ is at least

$$\frac{1}{4} \left( \sqrt{n(\kappa-1)} \right) \left( \ln \frac{\mu}{2} + 2\ln \frac{nR\mu}{\sqrt{2}(L-\mu)} + \ln(1/\epsilon) \right).$$

This, together with Theorem 5 below (through which we establish the $\Omega(n)$ part), concludes the proof.

∎

We bound from below the number of iterations required to obtain a non-trivial accuracy.

**Lemma 2.** *Let $j \in [n]$, let $\boldsymbol{\eta}_{j,1} \in \mathcal{H}$ be a parameters vector whose all coordinates are $-\frac{L-\mu}{2}$ and let $\boldsymbol{\eta}_{j,2} \in \mathcal{H}$ be a parameters vector whose all coordinates are $-\frac{L-\mu}{2}$, except for the $j$'th coordinate which we set to be $\frac{L-\mu}{2}$. If $\kappa > 3$, then*

$$\|\mathbf{w}^*(\boldsymbol{\eta}_1) - \mathbf{w}^*(\boldsymbol{\eta}_2)\| \geq \frac{2R}{n+2}.$$

**Proof** By Equation (13) we have

$$\|\mathbf{w} * (\boldsymbol{\eta}_1) - \mathbf{w} * (\boldsymbol{\eta}_2)\| = \sqrt{2} \left| \frac{R\mu}{\sqrt{2} \left( \frac{L+\mu}{2} + \frac{1}{n} \sum_{i=1}^{n} (\eta_1)_i \right)} - \frac{R\mu}{\sqrt{2} \left( \frac{L+\mu}{2} + \frac{1}{n} \sum_{i=1}^{n} (\eta_2)_i \right)} \right|$$

$$= R\mu \left| \frac{1}{\frac{L+\mu}{2} - \frac{L-\mu}{2}} - \frac{1}{\frac{L+\mu}{2} - \frac{(n-1)(L-\mu)}{2n} + \frac{L-\mu}{2n}} \right|$$

$$= R\mu \left| \frac{\frac{L+\mu}{2} - \frac{(n-1)(L-\mu)}{2n} + \frac{L-\mu}{2n} - \frac{L+\mu}{2} + \frac{L-\mu}{2}}{\left( \frac{L+\mu}{2} - \frac{L-\mu}{2} \right) \left( \frac{L+\mu}{2} - \frac{(n-1)(L-\mu)}{2n} + \frac{L-\mu}{2n} \right)} \right|$$

$$= R \left| \frac{-\frac{(n-1)(L-\mu)}{n} + \frac{L-\mu}{n} + L - \mu}{L + \mu - \frac{(n-1)(L-\mu)}{n} + \frac{L-\mu}{n}} \right|$$

$$= 2R \left| \frac{\frac{L-\mu}{n}}{L + \mu - \frac{(n-1)(L-\mu)}{n} + \frac{L-\mu}{n}} \right|$$

$$= 2R \left| \frac{1}{n\frac{\kappa+1}{\kappa-1} - (n-1) + 1} \right|$$

$$= 2R \left| \frac{1}{n\frac{\kappa+1}{\kappa-1} - n + 2} \right|$$

$$\geq \frac{2R}{n+2},$$

where the last inequality follows from $\kappa > 3$. ∎

**Theorem 5.** *The iteration complexity of any stochastic optimization algorithm (not necessarily CLI) which gathers information on $F_{\boldsymbol{\eta}}$ (with $\kappa > 3$) only by means of incremental oracles, i.e., oracles which upon receiving query return an answer which depends on not more than one individual function, is at least $n$.*

**Proof** Let $\mathcal{A}$ be a stochastic optimization algorithm. According to Yao's principle, we can bound from below the $\epsilon$-optimality of $\mathcal{A}$ after $k < n$ iterations by estimating the $\epsilon$-optimality of any deterministic algorithm w.r.t. to distribution $\mathcal{D}(\mathcal{H})$ over $\mathcal{H}$ defined by: draw $j \in [n]$ and set $\boldsymbol{\eta}$ to be $\boldsymbol{\eta}_{j,1}$ or $\boldsymbol{\eta}_{j,2}$ as defined in Lemma 2 w.p. $1/2$. Then,

$$\max_{\{\boldsymbol{\eta}_{j,i} | j \in [n], i \in [2]\}} \mathbb{E}_{\mathcal{A}}[F_{\boldsymbol{\eta}_{j,i}}(\mathbf{w}^{(k)}(\boldsymbol{\eta}_{j,i})) - F_{\boldsymbol{\eta}_{j,i}}(\mathbf{w}^*(\boldsymbol{\eta}_{j,i}))]$$

$$\geq \min_{\text{deterministic algorithms}} \mathbb{E}_{\boldsymbol{\eta} \sim \mathcal{D}(\mathcal{H})}[F_{\boldsymbol{\eta}}(\mathbf{w}^{(k)}(\boldsymbol{\eta}) - F_{\boldsymbol{\eta}}(\mathbf{w}^*(\boldsymbol{\eta}))]$$

$$\geq \min_{\text{deterministic algorithms}} \mathbb{E}_{\boldsymbol{\eta} \sim \mathcal{D}(\mathcal{H})} \frac{\mu}{2} \|\mathbf{w}^{(k)}(\boldsymbol{\eta}) - \mathbf{w}^*(\boldsymbol{\eta})\|^2$$

$$\geq \frac{\mu}{2} \min_{\text{deterministic algorithms}} \left( \mathbb{E}_{\boldsymbol{\eta} \sim \mathcal{D}(\mathcal{H})} \|\mathbf{w}^{(k)}(\boldsymbol{\eta}) - \mathbf{w}^*(\boldsymbol{\eta})\| \right)^2$$

$$\geq \frac{\mu}{2} \left( \frac{R}{n(n+2)} \right)^2,$$

where the last inequality follows from Lemma 2. Thus, for sufficiently small $\epsilon$, one must perform at least $n$ iterations in order to obtain an $\epsilon$-optimal solution. ∎

## A.3   Proof of Theorem 3 - Smooth Functions

The following notation

$$\overline{\mathcal{P}}_k := \{p \in \mathcal{P}_k | p(0) = 0\} \tag{25}$$

will come in handy in subsequent proofs.

**Lemma 3.** *When applied on*

$$g_\eta(\mathbf{x}) := \frac{\eta}{2} \|\mathbf{x}\|^2 - R\eta \mathbf{e}_1^\top \mathbf{x}, \quad \eta \in (0, L] \tag{26}$$

*with a first-order oracle (as defined in 10 with $n = 1$), the coordinates of iterates produced by oblivious stochastic CLIs whose is initialization iterate is $\mathbf{x}_i^{(0)} = 0$ form polynomials in $\eta$ with random real coefficients which vanishes at $\eta = 0$ and whose degree does not exceed the iteration number.*

**Proof** Let $\mathcal{A}$ be an oblivious stochastic CLI, and suppose we apply $\mathcal{A}$ on the class of problems (26) parameterized by $\eta$, using a first-order. We use mathematical induction to show that for any $k = 0, 1, \ldots$, the coordinate of the $k$'th iterate produced by such process can be expressed as a distribution over $\overline{\mathcal{P}}_k$.

As the first iterate $\mathbf{x}_i^{(0)}$ is assumed to be zero, the base case is trivial. For the inductive step, assume that any coordinate of $\mathbf{x}_i^{(k)}$ can be expressed as a distribution over $\overline{\mathcal{P}}_k$. It is easy to see that for any $\mathbf{x}_i^{(k)}$, the answers of the first-order oracle,

$$\text{First-order oracle:} \quad \mathcal{O}(\mathbf{x}_i^{(k)}; A, B, \mathbf{c}) = A(\eta \mathbf{x}_i^{(k)} - R\eta \mathbf{e}_1) + B\mathbf{x}_i^{(k)} + \mathbf{c},$$

form a distribution over $\mathcal{P}_{k+1}^0$, as the random quantities involved in the expressions ($A$, $B$ and $\mathbf{c}$) do not depend on $\eta$ (due to obliviousness) and the rest of the terms ($\eta$ and $R\eta \mathbf{e}_i$) are homogenous in $\eta$. Lastly, $\mathbf{x}_i^{(k+1)}$ are computed by simply summing up all the oracle answers, and as such, form again distributions over $\mathcal{P}_{k+1}^0$.

■

**Proof** [Theorem 3] Let $\mathcal{N}$ be an oblivious stochastic CLI and let $\alpha \in (-1, 0)$. Our derivation of lower bounds for stochastic CLIs is established via Yao principle. Fix some $k \in \{0, 1, \ldots\}$. By Lemma 3, $\mathbf{x}_1^{(k)}(\eta)$ distributes according to some distribution $\mathcal{D}$ over $(\overline{\mathcal{P}}_k)^d$. Thus, by Yao principle, since each vector of polynomials in $(\overline{\mathcal{P}}_k)^d$ represents a single deterministic algorithm, we have

$$\max_{\eta \in [0, L]} \mathbb{E}_{\mathbf{x}_1^{(k)}(\eta) \sim \mathcal{D}}[g_\eta(\mathbf{x}_1^{(k)}(\eta)) - g_\eta(\mathbf{x}^*(\eta))] \geq \min_{\mathbf{s}(\eta) \in (\overline{\mathcal{P}}_k)^d} \mathbb{E}_{\eta \sim \mathcal{E}([0,L])}[g_\eta(\mathbf{s}(\eta)) - g_\eta(\mathbf{x}^*(\eta))],$$

where $\mathcal{E}([0, L], \alpha)$ (abbr. $\mathcal{E}$) denotes a distribution over $(0, L]$ with a probability density function

$$p_\mathcal{E}(\eta) = \frac{(\alpha + 1)\eta^\alpha}{L^{\alpha+1}}.$$

We have,

$$\begin{aligned}
\min_{\mathbf{s}(\eta) \in (\overline{\mathcal{P}}_k)^d} \mathbb{E}_{\eta \sim \mathcal{E}}[g_\eta(\mathbf{s}(\eta)) - g_\eta(\mathbf{x}^*(\eta))] &\geq \min_{s(\eta) \in \overline{\mathcal{P}}_k} \mathbb{E}_{\eta \sim \mathcal{E}}\left[\eta \|\mathbf{s}(\eta) - \mathbf{x}^*(\eta)\|^2\right] \\
&\geq \min_{s(\eta) \in \overline{\mathcal{P}}_k} \mathbb{E}_{\eta \sim \mathcal{E}}\left[\eta(s(\eta) - R)^2\right] \\
&= R^2 \min_{s(\eta) \in \overline{\mathcal{P}}_k} \mathbb{E}_{\eta \sim \mathcal{E}}\left[\eta(s(\eta) - 1)^2\right] \\
&= \frac{R^2(\alpha + 1)}{L^{\alpha+1}} \min_{s(\eta) \in \overline{\mathcal{P}}_k} \int_0^L \eta(s(\eta) - 1)^2 \eta^\alpha d\eta \\
&= \frac{R^2(\alpha + 1)}{L^{\alpha+1}} \min_{s(\eta) \in \overline{\mathcal{P}}_k} \int_0^1 L\eta(s(L\eta) - 1)^2 (L\eta)^\alpha L \, d\eta \\
&= LR^2(\alpha + 1) \min_{s(\eta) \in \overline{\mathcal{P}}_k} \int_0^1 \eta(s(\eta) - 1)^2 \eta^\alpha \, d\eta
\end{aligned}$$

where the first inequality follows by the fact that $h_\eta$ is $\eta$-strongly convex and the second inequality follows by focusing on the first coordinate of $\mathbf{s}(\eta) - \mathbf{x}^*(\eta)$. Invoking Lemma 9 yields

$$LR^2(\alpha+1) \min_{s(\eta)\in\overline{\mathcal{P}}_k} \int_0^1 \eta(s(\eta)-1)^2\eta^\alpha \, d\eta = LR^2(\alpha+1) \min_{s(\eta)\in\mathcal{P}_{k-1}} \int_0^1 \eta(s(\eta)\eta-1)^2\eta^\alpha d\eta,$$

$$\geq \frac{LR^2(\alpha+1)}{e^2(k+2)^{2(\alpha+1)+2}}.$$

Thus, in this case the iteration complexity is bound from below by

$$\sqrt[2(\alpha+1)+2]{\frac{LR^2(\alpha+1)}{e^2\epsilon}} - 2.$$

∎

## A.4 Proof of Theorem 4 - Regularized Empirical Loss Minimization

For ease of presentation, we assume that $\|\mathbf{x}_i\| \leq 1$, $\phi_i$ take non-negative values and $\phi_i(0) \leq 1$. Furthermore, throughout the proof we assume that $n$ is even and that $L = 1$ (the proof for odd $n$ and general $L > 0$ holds mutatis mutandis). First, we give an explicit definition of the parametrized set of functions we will be focusing on, as well as the oracles under which our bounds hold. We denote by $\mathcal{H}$ the set of all $(\psi_1, \ldots, \psi_{n/2}) \in \mathbb{R}^{n/2}$ such that all the entries are 0, except for some $j \in [n/2]$, for which $\psi_j \in [-\pi/2, \pi/2]$. Now, given $\boldsymbol{\psi} \in \mathcal{H}$, we set

$$\phi_i(w) = \frac{1}{2}(w+1)^2 \implies \phi_i^*(u) = \frac{1}{2}u^2 - u$$

$$\mathbf{x}_{\boldsymbol{\psi},i} = \begin{cases} \cos(\psi_{(i+1)/2})\mathbf{e}_i + \sin(\psi_{(i+1)/2})\mathbf{e}_{i+1} & i \text{ is odd} \\ \mathbf{e}_i & \text{o.w.} \end{cases}.$$

In which case, the corresponding dual is:

$$D_{\boldsymbol{\psi}}(\boldsymbol{\alpha}) = \frac{1}{2n}\|\boldsymbol{\alpha}\|^2 - \frac{1}{n}\mathbf{1}^\top\boldsymbol{\alpha} + \frac{1}{2\lambda n^2}\|X_{\boldsymbol{\psi}}\alpha_i\|^2 \tag{27}$$

where

$$X_{\boldsymbol{\psi}} := (\mathbf{x}_{\boldsymbol{\psi},1}, \ldots, \mathbf{x}_{\boldsymbol{\psi},n}).$$

Equivalently

$$D_{\boldsymbol{\psi}} = \frac{1}{2}\boldsymbol{\alpha}^\top \left(\frac{1}{n}I + \frac{1}{\lambda n^2}X_{\boldsymbol{\psi}}^\top X_{\boldsymbol{\psi}}\right)\boldsymbol{\alpha} - \frac{1}{n}\mathbf{1}^\top\boldsymbol{\alpha}$$

Note that

$$Q_{\boldsymbol{\psi}} := \frac{1}{n}I + \frac{1}{\lambda n^2}X_{\boldsymbol{\psi}}^\top X_{\boldsymbol{\psi}} = \frac{1}{n}\begin{pmatrix} 1+\frac{1}{\lambda n} & \frac{1}{\lambda n}\sin(\psi_1) & & & \\ \frac{1}{\lambda n}\sin(\psi_1) & 1+\frac{1}{\lambda n} & & & \\ & & 1+\frac{1}{\lambda n} & \frac{1}{\lambda n}\sin(\psi_2) & \\ & & \frac{1}{\lambda n}\sin(\psi_2) & 1+\frac{1}{\lambda n} & \\ & & & & \ddots \end{pmatrix}.$$

Note that, all the eigenvalues of $Q_{\boldsymbol{\psi}}$ are bigger than 1. Therefore, $D_{\boldsymbol{\psi}}$ is 1-strongly convex. We assume that the oracles at the algorithms' disposal are the dual RLM oracles defined in (19),

Lastly, we will need the following definitions

$$\mathcal{P}_{k,d}^n(\eta_1, \eta_2, \ldots, \eta_n) := \left\{ \begin{pmatrix} p_1(\eta_1, \eta_2, \ldots, \eta_n) \\ \vdots \\ p_d(\eta_1, \eta_2, \ldots, \eta_n) \end{pmatrix} \middle| p_1, \ldots, p_d \in \mathcal{P}_k^n, \quad \partial p_1 + \cdots + \partial p_d \leq k \right\} \tag{28}$$

$$\mathcal{Q}_{k,d}^n(\psi_1, \psi_2, \ldots, \psi_n) := \left\{ \begin{pmatrix} p_1(\sin\psi_1, \sin\psi_2, \ldots, \sin\psi_n) \\ \vdots \\ p_d(\sin\psi_1, \sin\psi_2, \ldots, \sin\psi_n) \end{pmatrix} \middle| p_1, \ldots, p_d \in \mathcal{P}_{k,d}^n \right\} \tag{29}$$

to ease notation in subsequent proofs (where $\partial p$ denotes the total degree of $p$ and $\mathcal{P}_k^n$ is defined in (20)). Thus, $\mathcal{Q}_k^n$ contains $d$-dimensional vectors whose entries are $n$-multivariate polynomials expressions in $\sin \psi_1, \ldots, \sin \psi_n$, such that the sum of the degrees of the $d$-polynomials does not exceed $k$. In addition, given $\mathbf{t}(\boldsymbol{\psi}) \in \mathcal{Q}_{k,d}^n$ we define

$$
t_i(\psi_i) := \mathbf{t}\left(0, \ldots, 0, \underbrace{\psi_i}_{i\text{'th entry}}, 0, \ldots, 0\right), \quad \forall i \in [d].
$$

As usual, we start by stating the functional form assumed by iterates produced by this sort of optimization algorithms.

**Lemma 4.** *When applied on (27) with a dual RLM oracle (as defined in 19), the coordinates of iterates produced by oblivious stochastic CLIs form $n$ multivariate polynomials expressions in $\sin \psi_1, \ldots, \sin \psi_{n/2}$ with random coefficients, such that the sum of the degrees of these polynomials does not exceed the iteration number.*

**Proof** Let $\mathcal{A}$ be a oblivious stochastic CLI, and suppose we apply $\mathcal{A}$ on the class of problems (27) parameterized by $\boldsymbol{\psi}$, using dual RLM oracles as defined in 19. We use mathematical induction to show that for any $k = 0, 1, \ldots$, the coordinate of the $k$'th iterate produced by such process can be expressed as a distribution over polynomial expressions in $\sin \psi_1, \ldots, \sin \psi_{n/2}$ whose sum of degrees is less than or equal $k$.

As the first iterate $\boldsymbol{\alpha}_i^{(0)}$ is allowed to depend only on $n$ and $\lambda$, the base case is trivial. That is, $\boldsymbol{\alpha}_i^{(0)}$ forms a distribution over $\mathbb{R}^n = \mathcal{Q}_{0,n}^{n/2}$ which does not depend on $\sin \psi_1, \ldots, \sin \psi_{n/2}$.

For the inductive step, assume that $\boldsymbol{\alpha}_i^{(k)}$ can be expressed as a distribution over $\mathcal{Q}_{k,n}^{n/2}$. It is easy to see that for any $\boldsymbol{\alpha}_i^{(k)}$, the answer of the dual RLM oracle

$$
\mathcal{O}(\boldsymbol{\alpha}_i^{(k)}; t, \ell) = \boldsymbol{\alpha} + t\mathbf{e}_\ell^\top (Q_{\boldsymbol{\psi}} \boldsymbol{\alpha}_i^{(k)} - \frac{1}{n}\mathbf{1})\mathbf{e}_\ell, \quad t \in \mathbb{R}, j \in [n],
$$

$$
\mathcal{O}(\boldsymbol{\alpha}_i^{(k)}; \ell) = \left(I - \frac{1}{(Q_{\boldsymbol{\psi}})_{\ell\ell}}\mathbf{e}_\ell (Q_{\boldsymbol{\psi}})_{\ell,*}\right) \boldsymbol{\alpha}^{(k)} + \frac{1}{n(Q_{\boldsymbol{\psi}})_{\ell\ell}}\mathbf{e}_\ell
$$

are distributions over $\mathcal{Q}_{k+1,n}^{n/2}$, as the only random quantity involved in the expressions $t, \ell$ does not depend on $\boldsymbol{\psi}$ (due to obliviousness), the only linear factor in $\sin \psi_\ell$ (i.e., $\mathbf{e}_\ell^\top (Q_{\boldsymbol{\psi}}\boldsymbol{\alpha} - \frac{1}{n}\mathbf{1})\mathbf{e}_\ell, \mathbf{e}_\ell(Q_{j,\eta})_{\ell,*})$ 'touches' $\boldsymbol{\alpha}_i^{(k)}$ at exactly one entry and the rest of the terms $(1/n\mathbf{1}, I, 1/(Q_{j,\eta})_{\ell\ell}$ and $n)$ are constants (w.r.t. $\sin \psi_\ell$). Lastly, $\boldsymbol{\alpha}_i^{(k+1)}$ are computed by simply summing up all the oracle answers, and as such, form again distributions over $\mathcal{Q}_{k+1,n}^{n/2}$. ∎

**Proof** [Theorem 4]

Let $\mathcal{A}$ be a oblivious stochastic CLI. By Lemma 4 the coordinates of $\boldsymbol{\alpha}_1^{(k)}$ (the point returned by the algorithm at the $k$'th iteration) when applied on the class of problems (27) distributes according to some distribution $\mathcal{D}$ over $(\mathcal{Q}_k^{n/2})^n$. Furthermore, it is easy to verify that the corresponding minimizers of (27) are

$$
\boldsymbol{\alpha}^*(\boldsymbol{\psi}) = \left(\frac{1}{\frac{\lambda n+1}{\lambda n} + \frac{1}{\lambda n}\sin(\psi_1)}, \frac{1}{\frac{\lambda n+1}{\lambda n} + \frac{1}{\lambda n}\sin(\psi_1)}, \frac{1}{\frac{\lambda n+1}{\lambda n} + \frac{1}{\lambda n}\sin(\psi_2)}, \frac{1}{\frac{\lambda n+1}{\lambda n} + \frac{1}{\lambda n}\sin(\psi_2)}, \ldots\right).
$$
(30)

$\boldsymbol{\alpha}_1^{(k)}(\boldsymbol{\psi})$ distributes according to some distribution $\mathcal{D}$ over $\mathcal{Q}_{k,n}^{n/2}$. Thus, by Yao principle, since each polynomial in $\mathcal{Q}_{k,n}^{n/2}$ represents a single deterministic algorithm, we have

$$
\max_{\boldsymbol{\psi} \in \mathcal{H}} \mathbb{E}_{\boldsymbol{\alpha}_1^{(k)}(\boldsymbol{\psi}) \sim \mathcal{D}} \|\boldsymbol{\alpha}_1^{(k)}(\boldsymbol{\psi}) - \boldsymbol{\alpha}^*(\boldsymbol{\psi})\| \geq \min_{\mathbf{t}(\boldsymbol{\psi}) \in \mathcal{Q}_{k,n}^{n/2}} \mathbb{E}_{\boldsymbol{\psi} \sim \mathcal{U}(\mathcal{H})} \|\mathbf{t}(\boldsymbol{\psi}) - \boldsymbol{\alpha}^*(\boldsymbol{\psi})\| \quad (31)
$$

where $\mathcal{U}(\mathcal{H})$ denotes a distribution over $\mathcal{H}$ which corresponds to of first drawing $j \sim \mathcal{U}([n/2])$ at random, and then drawing $\psi_j$ according to distribution defined by the p.d.f. $p_{\psi_j}(\psi) = \cos(\psi)/2$ over $[-\pi/2, \pi/2]$ (for $i \neq j$ we set $\psi_i = 0$ ). We now have,

$$
\min_{\mathbf{t}(\boldsymbol{\psi}) \in \mathcal{Q}_{k,n}^{n/2}} \mathbb{E}_{\boldsymbol{\psi} \sim \mathcal{U}(\mathcal{H})} \| \mathbf{t}(\boldsymbol{\psi}) - \boldsymbol{\alpha}^*(\boldsymbol{\psi}) \|
$$

$$
= \min_{\mathbf{t}(\boldsymbol{\psi}) \in \mathcal{Q}_{k,n}^{n/2}} \mathbb{E}_{j \sim \mathcal{U}([n/2])} \mathbb{E}_{\psi_j \sim \mathcal{U}([-\pi/2, \pi/2])} \| \mathbf{t}(\boldsymbol{\psi}) - \boldsymbol{\alpha}^*(\boldsymbol{\psi}) \|
$$

$$
= \frac{2}{n} \sum_{j=1}^{n/2} \min_{\mathbf{t}(\boldsymbol{\psi}) \in \mathcal{Q}_{k,n}^{n/2}} \mathbb{E}_{\psi_j \sim \mathcal{U}([-\pi/2, \pi/2])} \| \mathbf{t}(\boldsymbol{\psi}) - \boldsymbol{\alpha}^*(\boldsymbol{\psi}) \|
$$

$$
\geq \frac{2}{n} \sum_{j=1}^{n/2} \min_{\mathbf{t}(\boldsymbol{\psi}) \in \mathcal{Q}_{k,n}^{n/2}} \mathbb{E}_{\psi_j \sim \mathcal{U}([-\pi/2, \pi/2])} \left| t_j(\psi_j) - \frac{1}{\frac{\lambda n + 1}{\lambda n} + \frac{1}{\lambda n} \sin(\psi_j)} \right|
$$

$$
\geq \frac{1}{n} \sum_{j=1}^{n/2} \min_{\mathbf{t}(\boldsymbol{\psi}) \in \mathcal{Q}_{k,n}^{n/2}} \int_{-\pi/2}^{\pi/2} \left| t_j(\psi_j) - \frac{1}{\frac{\lambda n + 1}{\lambda n} + \frac{1}{\lambda n} \sin(\psi_j)} \right| \cos \psi_j \, d\psi_j
$$

$$
= \frac{1}{n} \sum_{j=1}^{n/2} \min_{\mathbf{s}(\boldsymbol{\psi}) \in \mathcal{Q}_{k,n}^{n/2}} \int_{-1}^{1} \left| s_j(\eta_j) - \frac{1}{\frac{\lambda n + 1}{\lambda n} + \frac{1}{\lambda n} \eta_j} \right| d\eta_j
$$

$$
= \lambda \sum_{j=1}^{n/2} \min_{\mathbf{s}(\boldsymbol{\psi}) \in \mathcal{Q}_{k,n}^{n/2}} \int_{-1}^{1} \left| s_j(\eta_j) - \frac{1}{\lambda n + 1 + \eta_j} \right| d\eta_j \tag{32}
$$

where the first inequality follows by focusing on the $j$'th coordinate of $\mathbf{s}(\psi) - \boldsymbol{\alpha}^*(\psi)$ in each summand. Now, set $\alpha = 1 + \lambda n, L = 3, \mu = 1$ and note that

$$
\sqrt{\frac{2\alpha + L - \mu}{2\alpha + \mu - L}} = \sqrt{\frac{2\lambda n + 4}{2\lambda n}} = \sqrt{\frac{\lambda n + 2}{\lambda n}} = \sqrt{\frac{2}{\lambda n} + 1}
$$

Thus, by Lemma 8, using the same value for $\alpha$ and noting that $\alpha > 1 = (L - \mu)/2$ yields

$$
\int_{-1}^{1} \left| s_j(\eta_j) - \frac{1}{\lambda n + 1 + \eta_j} \right| d\eta_j \geq \left( \frac{\sqrt{\frac{2}{\lambda n} + 1} - 1}{\sqrt{\frac{2}{\lambda n} + 1} + 1} \right)^{k_j}
$$

where $k_j$ denotes the degree of $s_j(\eta_j)$. Plugging in this into Inequality (32) we get

$$
\max_{\boldsymbol{\psi} \in \mathcal{H}} \mathbb{E}_{\boldsymbol{\alpha}_1^{(k)}(\boldsymbol{\psi}) \sim \mathcal{D}} \| \boldsymbol{\alpha}_1^{(k)}(\boldsymbol{\psi}) - \boldsymbol{\alpha}^*(\boldsymbol{\psi}) \| \geq \lambda \min_{\mathbf{s}(\boldsymbol{\psi}) \in \mathcal{Q}_{k,n}^{n/2}} \sum_{j=1}^{n/2} \left( \frac{\sqrt{\frac{2}{\lambda n} + 1} - 1}{\sqrt{\frac{2}{\lambda n} + 1} + 1} \right)^{k_j} .
$$

Since $u \mapsto \rho^u$ is a decreasing and convex function for any $1 > \rho > 0$, we have

$$
\lambda \min_{\mathbf{s}(\boldsymbol{\psi}) \in \mathcal{Q}_{k,n}^{n/2}} \sum_{j=1}^{n/2} \left( \frac{\sqrt{\frac{2}{\lambda n} + 1} - 1}{\sqrt{\frac{2}{\lambda n} + 1} + 1} \right)^{k_j} \geq n\lambda/2 \min_{\mathbf{s}(\boldsymbol{\psi}) \in \mathcal{Q}_{k,n}^{n/2}} \left( \frac{\sqrt{\frac{2}{\lambda n} + 1} - 1}{\sqrt{\frac{2}{\lambda n} + 1} + 1} \right)^{\frac{2}{n} \sum_{j=1}^{n/2} k_j}
$$

$$
\geq n\lambda/2 \min_{\mathbf{s}(\boldsymbol{\psi}) \in \mathcal{Q}_{k,n}^{n/2}} \left( \frac{\sqrt{\frac{2}{\lambda n} + 1} - 1}{\sqrt{\frac{2}{\lambda n} + 1} + 1} \right)^{\frac{2k}{n}}
$$

where the last inequality is due to the fact that $\mathbf{s}(\boldsymbol{\psi}) \in \mathcal{Q}_{k,n}^{n/2}(\sin\psi)$ which implies that $\sum_{j=1}^{n} k_j \leq k$. Finally, we have,

$$\max_{\boldsymbol{\psi}\in\mathcal{H}}\mathbb{E}_{\boldsymbol{\alpha}_1^{(k)}(\boldsymbol{\psi})\sim\mathcal{D}}[D_{\boldsymbol{\psi}}(\boldsymbol{\alpha}_1^{(k)}(\boldsymbol{\psi})) - D_{\boldsymbol{\psi}}(\boldsymbol{\alpha}^*(\boldsymbol{\psi}))] \geq \max_{\boldsymbol{\psi}\in\mathcal{H}}\mathbb{E}_{\boldsymbol{\alpha}_1^{(k)}(\boldsymbol{\psi})\sim\mathcal{D}}\frac{1}{2}\|\boldsymbol{\alpha}_1^{(k)}(\boldsymbol{\psi}) - \boldsymbol{\alpha}^*(\boldsymbol{\psi})\|^2$$

$$\geq \frac{1}{2}\left(\max_{\boldsymbol{\psi}\in\mathcal{H}}\mathbb{E}_{\boldsymbol{\alpha}_1^{(k)}(\boldsymbol{\psi})\sim\mathcal{D}}\|\boldsymbol{\alpha}_1^{(k)}(\boldsymbol{\psi}) - \boldsymbol{\alpha}^*(\boldsymbol{\psi})\|\right)^2$$

$$\geq \frac{1}{2}\left(n\lambda/2 \min_{\mathbf{s}(\boldsymbol{\psi})\in\mathcal{Q}_{k,n}^{n/2}}\left(\frac{\sqrt{\frac{2}{\lambda n}+1}-1}{\sqrt{\frac{2}{\lambda n}+1}+1}\right)^{\frac{2k}{n}}\right)^2,$$

where the first inequality follows by the 1-strong convexity of $D_{\boldsymbol{\psi}}$ and the third inequality follows by Jensen inequality. Using Lemma 12, we get that the iteration complexity of $\mathcal{A}$ is at least

$$\frac{1}{8}\sqrt{\frac{2n}{\lambda}}\left(\ln\frac{n^2\lambda^2}{8} + \ln(1/\epsilon)\right).$$

∎

Lastly, we bound from below the number of iterations required to obtain a non-trivial accuracy.

**Lemma 5.** *Let $j \in [n]$, let $\boldsymbol{\psi}_{j,1} \in \mathcal{H}$ be a parameters vector whose all coordinates are $-\pi/2$ and let $\boldsymbol{\eta}_{\psi,2} \in \mathcal{H}$ be a parameters vector whose all coordinates are $-\pi/2$, except for the $j$'th coordinate which we set to be $\pi/2$. Then*

$$\|\boldsymbol{\alpha}^*(\boldsymbol{\psi}_1) - \boldsymbol{\alpha}^*(\boldsymbol{\psi}_2)\| \geq \frac{2\sqrt{2}}{\lambda n + 2}$$

**Proof** By Equation (30) we have

$$\|\boldsymbol{\alpha}^*(\boldsymbol{\psi}_1) - \boldsymbol{\alpha}^*(\boldsymbol{\psi}_2)\| = \sqrt{2}\left(\frac{1}{\frac{\lambda n+1}{\lambda n} - \frac{1}{\lambda n}} - \frac{1}{\frac{\lambda n+1}{\lambda n} + \frac{1}{\lambda n}}\right)$$

$$= \sqrt{2}\left(1 - \frac{\lambda n}{\lambda n + 2}\right)$$

$$= \frac{2\sqrt{2}}{\lambda n + 2}.$$

∎

**Theorem 6.** *When applied on (27), the iteration complexity of oblivious stochastic CLI algorithms equipped with a dual RLM oracle $D_{\boldsymbol{\psi}}$ is at least $n/2$.*

**Proof** Let $\mathcal{A}$ be a stochastic optimization algorithm. By Lemma 4 the coordinates of $\boldsymbol{\alpha}_1^{(k)}$ (the point returned by the algorithm at the $k$'th iteration) when applied on the class of problems (27) distributes according to some distribution $\mathcal{D}$ over $(\mathcal{Q}_k^{n/2})^n$. By Yao principle, since each polynomial in $\mathcal{Q}_{k,n}^{n/2}$ represents a single deterministic algorithm, we have

$$\max_{\boldsymbol{\psi}\in\mathcal{H}}\mathbb{E}_{\boldsymbol{\alpha}_1^{(k)}(\boldsymbol{\psi})\sim\mathcal{D}}\|\boldsymbol{\alpha}_1^{(k)}(\boldsymbol{\psi}) - \boldsymbol{\alpha}^*(\boldsymbol{\psi})\| \geq \min_{\mathbf{t}(\boldsymbol{\psi})\in\mathcal{Q}_{k,n}^{n/2}}\mathbb{E}_{\boldsymbol{\psi}\sim\mathcal{D}(\mathcal{H})}\|\mathbf{t}(\boldsymbol{\psi}) - \boldsymbol{\alpha}^*(\boldsymbol{\psi})\| \qquad (33)$$

where $\mathcal{D}(\mathcal{H})$ denotes a distribution over $\mathcal{H}$ which corresponds to the process of first drawing $j \sim \mathcal{U}([n/2])$ at random, and then set $\boldsymbol{\psi}$ to be $\boldsymbol{\psi}_{j,1}$ or $\boldsymbol{\psi}_{j,2}$ as defined in Lemma 5 with equal probability.

Now, for $k < n/2$, there exists some $j \in [n/2]$ such that $\mathbf{t}(\boldsymbol{\psi})$ does not depend on $\psi_j$. This yields,

$$\max_{\{\boldsymbol{\psi}_{j,i}|j\in[n/2],i\in[2]\}} \mathbb{E}_{\mathcal{A}}[D_{\boldsymbol{\psi}_{j,i}}(\boldsymbol{\alpha}^{(k)}(\boldsymbol{\psi}_{j,i})) - D_{\boldsymbol{\psi}_{j,i}}(\boldsymbol{\alpha}^{*}(\boldsymbol{\psi}_{j,i}))]$$

$$\geq \min_{\text{deterministic algorithms}} \mathbb{E}_{\boldsymbol{\psi}\sim\mathcal{D}(\mathcal{H})}[D_{\boldsymbol{\psi}_{j,i}}(\boldsymbol{\alpha}^{(k)}(\boldsymbol{\psi}_{j,i})) - D_{\boldsymbol{\psi}_{j,i}}(\boldsymbol{\alpha}^{*}(\boldsymbol{\psi}_{j,i}))]$$

$$\geq \min_{\text{deterministic algorithms}} \mathbb{E}_{\boldsymbol{\psi}\sim\mathcal{D}(\mathcal{H})}\frac{1}{2}\|\boldsymbol{\alpha}^{(k)}(\boldsymbol{\psi}_{j,i})) - \boldsymbol{\alpha}^{*}(\boldsymbol{\psi}_{j,i})\|^2$$

$$\geq \frac{1}{2}\min_{\text{deterministic algorithms}}\left(\mathbb{E}_{\boldsymbol{\psi}\sim\mathcal{D}(\mathcal{H})}\|\boldsymbol{\alpha}^{(k)}(\boldsymbol{\psi}_{j,i})) - \boldsymbol{\alpha}^{*}(\boldsymbol{\psi}_{j,i})\|\right)^2$$

$$\geq \frac{1}{2}\left(\frac{2\sqrt{2}}{n(\lambda n + 2)}\right)^2,$$

where the last inequality follows from Lemma 5. Thus, for sufficiently small $\epsilon$, one must perform at least $n/2$ iterations in order to obtain an $\epsilon$-optimal solution. ∎

### A.5 Best polynomial approximation over closed intervals in $\mathbb{R}$

In the following section we analyze the best polynomial approximation of some functions w.r.t. $L_\infty$, $L_1$ and $L_2$ norm, based on standard results from the approximation theory (see generally, Allan Pinkus. On L1-approximation, 1989; Theodore J Rivlin. An introduction to the approximation of functions, 2003; Ronald A DeVore and George G Lorentz. Constructive approximation, 1993; Naum Il'ich Akhiezer and Charles J Hyman. Theory of approximation. Translated by Charles J. Hyman. New York, 1956; Isidor Pavlovich Natanson. Constructive function theory, 1964).

#### A.5.1 Approximation w.r.t. $L_\infty$

**Lemma 6.** *Let $b > a > 0$ and $c > -a$, then*

$$\min_{s(\eta)\in\mathcal{P}_k}\left\|s(\eta) - \frac{1}{\eta + c}\right\|_{L_\infty([a,b])} \geq \frac{2(b-a)}{(b+a+2c)^2 - (b-a)^2}\left(\frac{\sqrt{\frac{b+c}{a+c}} - 1}{\sqrt{\frac{b+c}{a+c}} + 1}\right)^k.$$

**Proof** We have,

$$\min_{s(\eta)\in\mathcal{P}_k}\left\|s(\eta) - \frac{1}{\eta + c}\right\|_{L_\infty([a,b])} = \min_{s(\eta)\in\mathcal{P}_k}\left\|s\left(\frac{a-b}{2}\eta + \frac{a+b}{2}\right) - \frac{1}{\frac{a-b}{2}\eta + \frac{b+a}{2} + c}\right\|_{L_\infty([-1,1])}$$

$$= \min_{s(\eta)\in\mathcal{P}_k}\left\|s(\eta) - \frac{1}{\frac{a-b}{2}\eta + \frac{b+a+2c}{2}}\right\|_{L_\infty([-1,1])}$$

$$= \frac{2}{b-a}\min_{s(\eta)\in\mathcal{P}_k}\left\|\frac{b-a}{2}s(\eta) - \frac{\frac{b-a}{2}}{\frac{a-b}{2}\eta + \frac{b+a+2c}{2}}\right\|_{L_\infty([-1,1])}$$

$$= \frac{2}{b-a}\min_{s(\eta)\in\mathcal{P}_k}\left\|s(\eta) + \frac{1}{\eta - \frac{b+a+2c}{b-a}}\right\|_{L_\infty([-1,1])}$$

$$= \frac{2}{b-a}\min_{s(\eta)\in\mathcal{P}_k}\left\|-s(\eta) + \frac{1}{\eta - \frac{b+a+2c}{b-a}}\right\|_{L_\infty([-1,1])}$$

$$= \frac{2}{b-a}\min_{s(\eta)\in\mathcal{P}_k}\left\|s(\eta) - \frac{1}{\eta - \frac{b+a+2c}{b-a}}\right\|_{L_\infty([-1,1])}$$

where we used the fact that $\mathcal{P}_k$ is invariant under pre-composition and post-composition with linear function in the second, fourth and fifth equalities. Now, since

$$c > -a \implies \frac{b+a+2c}{b-a} > 1,$$

combining Inequality 8 with Lemma 10, yields

$$\min_{s(\eta)\in\mathcal{P}_k} \left\| s(\eta) - \frac{1}{\eta - \frac{b+a+2c}{b-a}} \right\|_{L_\infty([-1,1])} \geq \frac{\left( \left(\frac{b+a+2c}{b-a}\right) - \sqrt{\left(\frac{b+a+2c}{b-a}\right)^2 - 1} \right)^k}{\left(\frac{b+a+2c}{b-a}\right)^2 - 1}$$

$$= \frac{1}{\left(\frac{b+a+2c}{b-a}\right)^2 - 1} \left( \frac{1 - \sqrt{\frac{\frac{b+a+2c}{b-a}-1}{\frac{b+a+2c}{b-a}+1}}}{1 + \sqrt{\frac{\frac{b+a+2c}{b-a}-1}{\frac{b+a+2c}{b-a}+1}}} \right)^k \qquad (34)$$

Noting that

$$\frac{\frac{b+a+2c}{b-a} - 1}{\frac{b+a+2c}{b-a} + 1} = \frac{b+a+2c-(b-a)}{b+a+2c+b-a} = \frac{a+c}{b+c},$$

we get

$$\min_{s(\eta)\in\mathcal{P}_k} \left\| s(\eta) - \frac{1}{\eta + c} \right\|_{L_\infty([a,b])} \geq \frac{2}{b-a} \frac{1}{\left(\frac{b+a+2c}{b-a}\right)^2 - 1} \left( \frac{\sqrt{\frac{b+c}{a+c}} - 1}{\sqrt{\frac{b+c}{a+c}} + 1} \right)^k.$$

Finally, since

$$\frac{2}{b-a} \frac{1}{\left(\frac{b+a+2c}{b-a}\right)^2 - 1} = \frac{2}{b-a} \frac{(b-a)^2}{(b+a+2c)^2 - (b-a)^2} = \frac{2(b-a)}{(b+a+2c)^2 - (b-a)^2},$$

we get

$$\min_{s(\eta)\in\mathcal{P}_k} \left\| s(\eta) - \frac{1}{\eta + c} \right\|_{L_\infty([a,b])} \geq \frac{2(b-a)}{(b+a+2c)^2 - (b-a)^2} \left( \frac{\sqrt{\frac{b+c}{a+c}} - 1}{\sqrt{\frac{b+c}{a+c}} + 1} \right)^k.$$

∎

### A.5.2 Approximation w.r.t. $L_1$

Let $U_k(\eta)$ denote the $k$'th second order Chebyshev polynomial, i.e.,

$$U_k(\eta) := \frac{\sin((k+1)\arccos\eta)}{\sqrt{1-\eta^2}} \qquad (35)$$

(To see why these are indeed polynomials, observe that $U_k(\eta)$ are the derivative of Chebyshev polynomials of first order scaled by a factor of $1/k$. The zeros of $U_k(\eta)$ are $\eta_j = \cos(\frac{j\pi}{k+1})$, $j = 1, \ldots, k$. First, let us establish the orthogonality of $\text{sgn}(U_k(\eta))$ with respect to $\mathcal{P}_{k-1}$ over $[-1, 1]$.

**Lemma 7.** *Let $p(\eta) \in \mathcal{P}_{k-1}$, then*

$$\int_{-1}^{1} p(\eta)\text{sgn}(U_k(\eta))\, d\eta = 0$$

**Proof** We integrate by substituting $\eta = \frac{e^{i\theta}+e^{-i\theta}}{2}$ ($=\cos(\theta)$),

$$\int_{-1}^{1} p(\eta)\mathrm{sgn}(U_k(\eta))\, d\eta \overset{\eta=\frac{e^{i\theta}+e^{-i\theta}}{2}}{=} \int_{\pi}^{0} p\left(\frac{e^{i\theta}+e^{-i\theta}}{2}\right)\mathrm{sgn}\left(U_k\left(\frac{e^{i\theta}+e^{-i\theta}}{2}\right)\right)\left(\frac{ie^{i\theta}-ie^{-i\theta}}{2}\right)\, d\theta$$

$$= \int_{0}^{\pi} p\left(\frac{e^{i\theta}+e^{-i\theta}}{2}\right)\mathrm{sgn}\left(\frac{\sin((k+1)\theta)}{\sin(\theta)}\right)\left(\frac{ie^{-i\theta}-ie^{i\theta}}{2}\right)\, d\theta$$

$$= \int_{0}^{\pi} p\left(\frac{e^{i\theta}+e^{-i\theta}}{2}\right)\mathrm{sgn}\left(\sin((k+1)\theta)\right)\left(\frac{ie^{-i\theta}-ie^{i\theta}}{2}\right)\, d\theta$$

$$= \frac{1}{2}\int_{-\pi}^{\pi} p\left(\frac{e^{i\theta}+e^{-i\theta}}{2}\right)\mathrm{sgn}\left(\sin((k+1)\theta)\right)\left(\frac{ie^{-i\theta}-ie^{i\theta}}{2}\right)\, d\theta \quad (36)$$

where the last equality is due to the fact that the integrand is an even function in $\theta$. Lastly, since for any $j = 1, \ldots, k$ we have

$$\int_{-\pi}^{\pi} e^{-ij\theta}\mathrm{sgn}(\sin((k+1)\theta))d\theta = \int_{-\pi+2\pi/(k+1)}^{\pi+2\pi/(k+1)} e^{-ij\theta}\mathrm{sgn}(\sin((k+1)\theta))d\theta$$

$$= \int_{-\pi}^{\pi} e^{-ij(\theta+2\pi/(k+1))}\mathrm{sgn}(\sin((k+1))(\theta+2\pi/(k+1)))d\theta$$

$$= e^{-2\pi ij/(k+1)}\int_{-\pi}^{\pi} e^{-ij\theta}\mathrm{sgn}(\sin((k+1)\theta))d\theta,$$

and since $e^{-2\pi ij/(k+1)} \neq 1$ for $j = 1, \ldots, k$, it follows that

$$\int_{-\pi}^{\pi} e^{-ij\theta}\sin((k+1)\theta)d\theta = 0, \quad j = 1, \ldots, k.$$

This, together with the case where $j = 0$,

$$\int_{-\pi}^{\pi} \sin((k+1)\theta)d\theta = \left(-\cos((k+1)\theta)/(k+1)\right)\Big|_{-\pi}^{\pi} = 0,$$

implies that all the terms in (36) vanish, thus concluding the proof. ∎

Given $\mu < L$ (note that, here $\mu$ and $L$ are allowed to take negative values), we define

$$\tilde{U}_k(\eta) := U_k\left(\frac{2\eta}{\mu - L}\right).$$

By substituting $\eta$ for $\frac{(\mu-L)\eta}{2}$, we get the following corollary.

**Corollary 1.** *Let* $p(\eta) \in \mathcal{P}_{k-1}$, *then*

$$\int_{-\frac{L-\mu}{2}}^{\frac{L-\mu}{2}} p(\eta)\,\mathrm{sgn}(\tilde{U}_k(\eta))\, d\eta = 0 \quad (37)$$

We now use Corollary 1 to bound from below the best polynomial $L_1$-approximation error w.r.t. $1/(\eta + \alpha)$ over the interval $[\mu, L]$.

**Lemma 8.** *Let* $p(\eta) \in \mathcal{P}_{k-1}$. *Then, for any* $(L - \mu)/2 < \alpha$ *we have*

$$\int_{-\frac{L-\mu}{2}}^{\frac{L-\mu}{2}} |p(\eta) - 1/(\eta + \alpha)|d\eta \geq \left(\frac{\sqrt{\frac{2\alpha+L-\mu}{2\alpha+\mu-L}} - 1}{\sqrt{\frac{2\alpha+L-\mu}{2\alpha+\mu-L}} + 1}\right)^{k}.$$

**Proof** First, note that the following two inequalities

$$\int_{-\frac{L-\mu}{2}}^{\frac{L-\mu}{2}} |p(\eta) - 1/(\eta + \alpha)|d\eta \geq \int_{-\frac{L-\mu}{2}}^{\frac{L-\mu}{2}} (p(\eta) - 1/(\eta + \alpha))\mathrm{sgn}(\tilde{U}_k(\eta))d\eta = -\int_{-\frac{L-\mu}{2}}^{\frac{L-\mu}{2}} 1/(\eta + \alpha)\mathrm{sgn}(\tilde{U}_k(\eta))d\eta$$

$$\int_{-\frac{L-\mu}{2}}^{\frac{L-\mu}{2}} |p(\eta) - 1/(\eta + \alpha)|d\eta \geq \int_{-\frac{L-\mu}{2}}^{\frac{L-\mu}{2}} (p(\eta) - 1/(\eta + \alpha))\mathrm{sgn}(-\tilde{U}_k(\eta))d\eta = \int_{-\frac{L-\mu}{2}}^{\frac{L-\mu}{2}} 1/(\eta + \alpha)\mathrm{sgn}(\tilde{U}_k(\eta))d\eta$$

hold due to orthogonality condition (37) and the fact that $\mathrm{sgn}(\cdot)$ is odd. Therefore,

$$\int_{-\frac{L-\mu}{2}}^{\frac{L-\mu}{2}} |p(\eta) - 1/(\eta + \alpha)| d\eta \geq \left| \int_{-\frac{L-\mu}{2}}^{\frac{L-\mu}{2}} 1/(\eta + \alpha) \, \mathrm{sgn}(\tilde{U}_k(\eta)) d\eta \right|.$$

Substituting $\frac{\mu - L}{2}\eta$ for $\eta$, yields

$$\left| \int_{-\frac{L-\mu}{2}}^{\frac{L-\mu}{2}} 1/(\eta + \alpha) \, \mathrm{sgn}(\tilde{U}_k(\eta)) d\eta \right| = \left| \int_{1}^{-1} \frac{1}{\frac{\mu - L}{2}\eta + \alpha} \, \mathrm{sgn}(U_k(\eta)) \frac{\mu - L}{2} d\eta \right|$$

$$= \left| \int_{-1}^{1} \frac{1}{\frac{\mu - L}{2}\eta + \alpha} \, \mathrm{sgn}(U_k(\eta)) \frac{L - \mu}{2} d\eta \right|$$

$$= \left| \int_{-1}^{1} \frac{1}{-\eta + \frac{2\alpha}{L - \mu}} \, \mathrm{sgn}(U_k(\eta)) d\eta \right| \qquad (38)$$

Now, plugging in the definition of $U_k(\eta)$ (see (35)) and applying Lemma 11, we get

$$\int_{-1}^{1} \frac{\mathrm{sgn}(\sin(k \arccos(\eta)))}{u - \eta} d\eta \geq \left( \frac{1 - \sqrt{\frac{u-1}{u+1}}}{1 + \sqrt{\frac{u-1}{u+1}}} \right)^k$$

for any $u > 1$. Using this inequality with (38) where $u = \frac{2\alpha}{L - \mu}$, yields

$$\int_{-\frac{L-\mu}{2}}^{\frac{L-\mu}{2}} |p(\eta) - 1/(\eta + \alpha)| d\eta \geq \left( \frac{\sqrt{\frac{2\alpha + L - \mu}{2\alpha + \mu - L}} - 1}{\sqrt{\frac{2\alpha + L - \mu}{2\alpha + \mu - L}} + 1} \right)^k.$$

■

### A.5.3 Approximation w.r.t. $L_2$

**Lemma 9.** *For any $\alpha \in (-1, 0)$,*

$$\min_{s(\eta) \in \mathcal{P}_{k-1}} \int_0^1 \eta(s(\eta)\eta - 1)^2 \eta^\alpha d\eta \geq \frac{1}{e^2(k+2)^{2(\alpha+1)+2}}$$

**Proof** Rephrasing it equivalently as

$$\min_{s(\eta) \in \mathcal{P}_{k-1}} \int_0^1 (s(\eta)\eta^{\frac{3+\alpha}{2}} - \eta^{\frac{1+\alpha}{2}})^2 d\eta,$$

shows that this problem can be seen as a best $L_2$-approximation for $\eta^{\frac{1+\alpha}{2}}$ in the $k$-dimensional space spanned by $g_i = \eta^{i + \frac{1+\alpha}{2}}$, $i = 1, \ldots, k$ (accordingly, $g_0 = \eta^{\frac{1+\alpha}{2}}$). By [2, Equation (3), p. 16], we have

$$\min_{s(\eta) \in \mathcal{P}_{k-1}} \int_0^1 (s(\eta)\eta^{\frac{3+\alpha}{2}} - \eta^{\frac{1+\alpha}{2}})^2 d\eta = \frac{\det G(g_0, g_1, \ldots, g_n)}{\det G(g_1, \ldots, g_n)}$$

where $G(\cdot)$ is Gram matrix (whose entries are the inner products of its arguments). First, note that

$$\langle g_i, g_j \rangle = \int_0^1 \eta^{i + \frac{1+\alpha}{2}} \eta^{j + \frac{1+\alpha}{2}} d\eta = \int_0^1 \eta^{i+j+1+\alpha} d\eta = \frac{1}{i + j + \alpha + 2}, \quad i, j = 0, 1, \ldots, k$$

Thus,

$$G(g_1, \ldots, g_k)_{i,j} = \frac{1}{i + j + \alpha + 2},$$

$$G(g_0, \ldots, g_k)_{i,j} = \frac{1}{i + j + \alpha}.$$

It follows that both matrices can be expressed as a Cauchy matrices, that is

$$G(g_1, \ldots, g_k)_{i,j} = \frac{1}{x_i - y_j},$$

$$G(g_0, \ldots, g_k)_{i,j} = \frac{1}{u_i - v_j}.$$

where $x_i = i + \alpha + 1, y_j = -j - 1$, $i, j \in [k]$ and $u_i = i + \alpha, v_j = -j$, $i, j \in [k+1]$. The determinant of Cauchy matrix $A$ defined by sequences $w_i, z_j$ one has

$$\det A = \frac{\prod_{i=2}^{k} \prod_{j=1}^{i-1} (w_i - w_j)(z_j - z_i)}{\prod_{i=1}^{k} \prod_{j=1}^{k} (w_i - z_j)}$$

Hence,

$$\frac{\det G(g_0, g_1, \ldots, g_k)}{\det G(g_1, \ldots, g_k)} = \frac{\frac{\prod_{i=2}^{k+1} \prod_{j=1}^{i-1} (u_i - u_j)(v_j - v_i)}{\prod_{i=1}^{k+1} \prod_{j=1}^{k+1} (u_i - v_j)}}{\frac{\prod_{i=2}^{k} \prod_{j=1}^{i-1} (x_i - x_j)(y_j - y_i)}{\prod_{i=1}^{k} \prod_{j=1}^{k} (x_i - y_j)}}$$

$$= \frac{\frac{\prod_{i=2}^{k+1} \prod_{j=1}^{i-1} (i-j)(-j-(-i))}{\prod_{i=1}^{k+1} \prod_{j=1}^{k+1} (i+\alpha-(-j))}}{\frac{\prod_{i=2}^{k} \prod_{j=1}^{i-1} ((i+1)-(j+1))((-j-1)-(-i-1))}{\prod_{i=1}^{k} \prod_{j=1}^{k} ((i+\alpha+1)-(-j-1))}}$$

$$= \frac{\frac{\prod_{i=2}^{k+1} \prod_{j=1}^{i-1} (i-j)^2}{\prod_{i=1}^{k+1} \prod_{j=1}^{k+1} (i+j+\alpha)}}{\frac{\prod_{i=2}^{k} \prod_{j=1}^{i-1} (i-j)^2}{\prod_{i=1}^{k} \prod_{j=1}^{k} (i+j+\alpha+2)}}$$

$$= \frac{\prod_{i=2}^{k+1} \prod_{j=1}^{i-1} (i-j)^2}{\prod_{i=1}^{k+1} \prod_{j=1}^{k+1} (i+j+\alpha)} \frac{\prod_{i=1}^{k} \prod_{j=1}^{k} (i+j+\alpha+2)}{\prod_{i=2}^{k} \prod_{j=1}^{i-1} (i-j)^2}$$

$$= \frac{\prod_{j=1}^{k} (k+1-j)^2 \prod_{i=1}^{k} \prod_{j=1}^{k} (i+j+\alpha+2)}{\prod_{i=1}^{k+1} \prod_{j=1}^{k+1} (i+j+\alpha)}$$

$$= \frac{\prod_{j=1}^{k} (k+1-j)^2 \prod_{i=1}^{k} \prod_{j=1}^{k} ((i+\alpha+1)+(j+1))}{\prod_{i=1}^{k+1} \prod_{j=1}^{k+1} ((i+\alpha)+j)}$$

$$= \frac{\prod_{j=1}^{k} (k+1-j)^2 \prod_{i=2}^{k+1} \prod_{j=2}^{k+1} ((i+\alpha)+j)}{\prod_{i=1}^{k+1} \prod_{j=1}^{k+1} ((i+\alpha)+j)}$$

$$= \frac{\prod_{j=1}^{k} (k+1-j)^2}{\prod_{i=1}^{k+1} (i+\alpha+1) \prod_{j=2}^{k+1} (1+\alpha+j)}$$

$$= \frac{\prod_{j=1}^{k} j^2}{\prod_{i=1}^{k+1} (i+\alpha+1) \prod_{j=2}^{k+1} (1+\alpha+j)}$$

$$= \frac{(\alpha+2) \prod_{j=1}^{k} j^2}{\prod_{i=1}^{k+1} (i+\alpha+1)^2}$$

$$= (\alpha+2) \left( \prod_{j=1}^{k} \frac{j}{j+\alpha+1} \right)^2 \frac{1}{(k+\alpha+2)^2}$$

To estimate the middle term, we apply arguments similar to the integral test for infinite series. First, note that,

$$\prod_{j=1}^{k} \frac{j}{j+\alpha+1} = \exp\left( \sum_{i=1}^{k} \ln \frac{j}{j+\alpha+1} \right).$$

Now, since for any $\alpha \in (-1, 0)$, it holds that $x \mapsto \ln \frac{x}{x+\alpha+1}$ is a monotone decreasing function (over $x \neq \alpha$), it holds that

$$\sum_{i=1}^{k} \ln \frac{j}{j+\alpha+1} \geq \int_{1}^{k+1} \ln \frac{x}{x+\alpha+1} dx = \left( x \ln \frac{x}{\alpha+x+1} - (\alpha+1)\ln(\alpha+x+1) \right)\Big|_{1}^{k+1}$$

Hence,

$$\prod_{j=1}^{k} \frac{j}{j+\alpha+1} \geq \exp\left( (k+1)\ln \frac{k+1}{\alpha+(k+1)+1} - (\alpha+1)\ln(\alpha+(k+1)+1) - \ln \frac{1}{\alpha+2} + (\alpha+1)\ln(\alpha+2) \right)$$

$$= \left( \frac{k+1}{k+\alpha+2} \right)^{k+1} (k+\alpha+2)^{-(\alpha+1)}(\alpha+2)^{\alpha+2}$$

$$\geq \left( \frac{k+1}{k+\alpha+2} \right)^{k+1} (k+\alpha+2)^{-(\alpha+1)}$$

$$= \left( 1 - \frac{\alpha+1}{k+\alpha+2} \right)^{k+1} (k+\alpha+2)^{-(\alpha+1)}$$

$$= \left( 1 - \frac{1}{\frac{k+1}{\alpha+1}+1} \right)^{k+1} (k+\alpha+2)^{-(\alpha+1)}.$$

Now, by the following standard inequality

$$1 - \frac{2}{x+1} \geq \exp\left( \frac{-2}{x-1} \right),$$

we get,

$$1 - \frac{1}{\frac{k+1}{\alpha+1}+1} = 1 - \frac{2}{(2\frac{k+1}{\alpha+1}+1)+1} \geq \exp\left( \frac{-2}{(2\frac{k+1}{\alpha+1}+1)-1} \right) = \exp\left( \frac{-1}{\frac{k+1}{\alpha+1}} \right) = \exp\left( \frac{-(\alpha+1)}{k+1} \right)$$

therefore,

$$(\alpha+2)\left( \prod_{j=1}^{k} \frac{j}{j+\alpha+1} \right)^2 \frac{1}{(k+\alpha+2)^2} \geq (\alpha+2)\left( \exp\left( \frac{-(\alpha+1)}{k+1} \right)^{k+1} (k+\alpha+2)^{-(\alpha+1)} \right)^2 \frac{1}{(k+\alpha+2)^2}$$

$$= (\alpha+2)\exp\left( -2(\alpha+1) \right)(k+\alpha+2)^{-2(\alpha+1)-2}$$

$$\geq (\alpha+2)\exp\left( -2(\alpha+1) \right)(k+2)^{-2(\alpha+1)-2}.$$

Lastly, since

$$(\alpha+2)\exp(-2(\alpha+1)) \geq \exp(-2),$$

for any $\alpha \in (-1, 0)$, we get

$$\min_{s(\eta)\in\mathcal{P}_{k-1}} \int_0^1 \eta(s(\eta)\eta - 1)^2 \eta^\alpha d\eta \geq \frac{1}{e^2(k+2)^{2(\alpha+1)+2}}.$$

■

## A.6 Technical Lemmas

**Lemma 10.** *For any $u \geq 1$,*

$$u - \sqrt{u^2 - 1} = \frac{1 - \sqrt{\frac{u-1}{u+1}}}{1 + \sqrt{\frac{u-1}{u+1}}}.$$

**Proof** We have,

$$\frac{1-\sqrt{\frac{u-1}{u+1}}}{1+\sqrt{\frac{u-1}{u+1}}} = \frac{\left(1-\sqrt{\frac{u-1}{u+1}}\right)^2}{1-\frac{u-1}{u+1}} = \frac{(u+1)\left(1-\sqrt{\frac{u-1}{u+1}}\right)^2}{u+1-(u-1)} = \frac{\left(\sqrt{u+1}-\sqrt{u-1}\right)^2}{2}$$

$$= \frac{u+1-2\sqrt{(u+1)(u-1)}+(u-1)}{2} = u - \sqrt{u^2-1}$$

∎

**Lemma 11.** *For any $u > 1$,*

$$\int_{-1}^{1} \frac{\operatorname{sgn}(\sin(k\arccos(\eta)))}{u-\eta}\,d\eta \geq \left(\frac{1-\sqrt{\frac{u-1}{u+1}}}{1+\sqrt{\frac{u-1}{u+1}}}\right)^k.$$

**Proof** First, note that the function

$$\gamma(x) := \ln\frac{x+1}{x-1} - \frac{1}{x}$$

takes non-negative for any $x > 1$, as

$$\gamma'(x) = \frac{x-1}{x+1}\frac{x-1-(x+1)}{(x-1)^2} + \frac{1}{x^2} = \frac{x-1}{x+1}\frac{-2}{(x-1)^2} + \frac{1}{x^2}$$

$$= \frac{-2}{(x+1)(x-1)} + \frac{1}{x^2} = \frac{-2}{x^2-1} + \frac{1}{x^2}$$

$$\leq \frac{-2}{x^2-1} + \frac{1}{x^2-1} = \frac{-1}{x^2-1} < 0$$

and $\lim_{x\to\infty}\gamma(x) = 0$. Therefore, by using identity (see Section F.31. in [2]), we get

$$\int_{-1}^{1}\frac{\operatorname{sgn}(\sin(k\arccos(\eta)))}{u-\eta}\,d\eta = 2\ln\frac{(u+\sqrt{u^2-1})^k+1}{(u+\sqrt{u^2-1})^k-1}$$

$$\geq (u-\sqrt{u^2-1})^k = \left(\frac{1-\sqrt{\frac{u-1}{u+1}}}{1+\sqrt{\frac{u-1}{u+1}}}\right)^k,$$

where the last equality is due to Lemma 10.

∎

**Lemma 12.** *Let $L > \mu > 0$, $c > 0$ and $\alpha \geq 0$. Then*

$$\epsilon \geq c\left(\frac{\sqrt{\frac{L+\alpha}{\mu+\alpha}}-1}{\sqrt{\frac{L+\alpha}{\mu+\alpha}}+1}\right)^k \implies k \geq \frac{1}{2}\left(\sqrt{\frac{L+\alpha}{\mu+\alpha}}-1\right)(\ln(c)+\ln(1/\epsilon))$$

**Proof** Note that the function

$$\delta(x) = \ln\frac{\sqrt{x}-1}{\sqrt{x}+1} + \frac{2}{\sqrt{x}-1}$$

takes non-negative values for $x > 1$, as

$$\delta'(x) = \frac{\sqrt{x}+1}{\sqrt{x}-1}\frac{0.5x^{-1/2}(\sqrt{x}+1)-0.5x^{-1/2}(\sqrt{x}-1)}{(\sqrt{x}+1)^2} - \frac{1}{(x-1)\sqrt{x-1}}$$

$$= \frac{1}{(x-1)\sqrt{x}} - \frac{1}{(x-1)\sqrt{x-1}} < 0$$

and $\lim_{x \to \infty} \delta(x) = 0$. Thus, we obtained the following inequality

$$\frac{\sqrt{x} - 1}{\sqrt{x} + 1} \geq \exp\left(\frac{-2}{\sqrt{x} - 1}\right), \ x > 1,$$

yields

$$c \left(\frac{\sqrt{\frac{L+\alpha}{\mu+\alpha}} - 1}{\sqrt{\frac{L+\alpha}{\mu+\alpha}} + 1}\right)^k \geq c \exp\left(\frac{-2k}{\sqrt{\frac{L+\alpha}{\mu+\alpha}} - 1}\right).$$

Hence,

$$\ln \epsilon \geq \ln(c) + \frac{-2k}{\sqrt{\frac{L+\alpha}{\mu+\alpha}} - 1}$$

$$\implies \frac{2k}{\sqrt{\frac{L+\alpha}{\mu+\alpha}} - 1} \geq \ln(c) + \ln(1/\epsilon)$$

$$\implies k \geq \frac{1}{2}\left(\sqrt{\frac{L+\alpha}{\mu+\alpha}} - 1\right)(\ln(c) + \ln(1/\epsilon))$$

∎