[Reviews · NeurIPS 2016]

Reviewer 1

Summary

This paper provides lower bounds for convex optimization problems with a finite sum structure. This is an important problem in machine learning as lower bounds can provide some insights to develop new optimization methods. Instead of relying on the oracle model developed by Nemirovsky et al., the authors extend the framework of Arjevani et al. [3, 5] that developed a structural approach in which new iterates are assumed to form linear combinations of the previous iterates and gradients. By reducing the problem of minimizing the considered objective function to a polynomial approximation problem, one can rely on Chebyshev polynomials to provide lower bounds.

Qualitative Assessment

Technical quality: The proofs derived in the paper are sound and well presented. One of the most interesting contributions is the lower bound for stochastic methods (including Stochastic Gradient Descent) which uses Yao's minimax principle, a neat and simple trick. The paper also provides some new insights, e.g. demonstrating that oblivious algorithms are bound to a linear rate as well as quantify the potential gain of various sampling schemes. Novelty/originality: Although the lower-bounds derived in this paper are of significant interest, I nevertheless have some concern with the current way the paper is written, especially concerning the differences to [5] that are not clearly stated in the paper. Although the authors seem to imply that they are the first one to derive dimension-free bounds, the work of [5] already derived lower bounds that hold independently of the dimension. My understanding is that the bounds of [5] apply for the setting where the number of iterations is greater than O(d/n). Can the authors elaborate on these aspects? I think the paper in its current state is rather unclear on this. I also would like to have the authors if there is any particular reason behind the choice of the quadratic function defined in Eq 12 which differs from the one used in [5]. I would like the authors to address the issue of the overlap with [5] before I consider updating my ratings. Minor: - Typos in Eq. 5 and appendix: should use variable $w$ instead of $x$. - Reference [1], please cite the ICML paper instead of arxiv. - Figure 1: “based on the function used by Nesterov”: please quote the function.

Confidence in this Review

2-Confident (read it all; understood it all reasonably well)


Reviewer 2

Summary

The authors provide a stronger lower bound than the one presented in Lan and Zhou (2015) in the sense that it covers a wider class of algorithms and/or problems.

Qualitative Assessment

The authors are suggested to clarify the following issues. 1) The lower bound in Lan and Zhou seems to be larger (and hence slightly stronger) than the one in Theorem 2. In particular, the bound obtained by Lan and Zhou [10] was (\sqrt{n \kappa} + n) \log (1/\epsilon) while the one in Theorem 2 was \sqrt{n \kappa} \log (1/\epsilon) +n. 2) An upper complexity bound was also obtained in [10] which matches the slightly stronger lower complexity bound, and the algorithm does not require the information about the optimal value, which might be an advantage over a proximal-point-like acceleration of SAG [11].

Confidence in this Review

2-Confident (read it all; understood it all reasonably well)


Reviewer 3

Summary

This paper provides lower bounds for iteration complexity of finite sum minimization and shows novel lower bounds that are valid for larger classes of algorithms. I think the obtained bounds complements the recent progresses that has been made in obtaining linear convergence rates for finite sum optimization with different algorithms and fills in the gap.

Qualitative Assessment

In contrast to existing lower bounds that are mostly information-theoretic, this paper derives the bounds using tools from approximation theory which is interesting in its own. However, the proposed techniques seems to basically extend the tools from an earlier work to finite sums, and mostly uses the analysis techniques developed in [5] (the authors are encouraged to honestly state the differences and key novelty of the present paper clearly).

Confidence in this Review

2-Confident (read it all; understood it all reasonably well)


Reviewer 4

Summary

Overall, this is a very nice paper which proposes lower bounds for iteration complexity of optimization algorithms based on approximation theory rather than information theory. Excellent intuition is given into the way arguments are constructed. This seems highly non trivial, yet accessibly communicated. They recover some known bounds, and show these bounds are valid for larger classes of algorithms than shown before using a single proof. The paper makes valuable points for the machine learning community along the way. After rebuttal: The authors did not provide feedback following the reviews, despite some (admittedly mostly mild) questions which have been raised.

Qualitative Assessment

In definition 2, Why allow for possibly more than one initialization point? Do you pay for that in some way? It doesn't seem so, judging by Definition 3 somewhat below. Or perhaps this is to allow for accelerated algorithms, which evolve more than one sequence simultaneously? The different sequences talk to each other. But somehow this extra cost of maintaining more than one sequence should be taken into account. If this is already the case, please state it explicitly. Section 2.3 is a great example of how this paper can convey clear messages about sophisticated proof techniques. Figure 2: is the x-axis \eta? Please label the axes. 3rd line after eq. (2): apply -> applies typeset equation after that: what is delta? line 72: latter -> later section 2.1: recall what L and µ are? What is MCG? Modified Conjugate Gradient? Method of Center of Gravity? line 125: smooth -> smoothness line 155: our -> out line 180: wrt to -> wrt equation 10: C should be a vector, not a matrix, right? line 196: Do you mean A = e_i e_i^T or e_ie_j^T ? If so, might as well be explicit, it's even shorter. line 210: obliviousness has 200 times more hits on google than obliviosity .. consider changing here and elsewhere. 245: The the

Confidence in this Review

2-Confident (read it all; understood it all reasonably well)


Reviewer 5

Summary

The authors provide lower bounds for smooth finite sum optimization which apply regardless of the dimension of the problem, and which match upper bounds given by certain iterative optimization methods like SDCA, SVRG, SAG, etc. The authors begin with a discussion of the limitations of many established lower bounds for finite sum optimization - namely that they apply only when the number of iterations is less than d/n where d is the dimension of the problem and n is the number of components in the sum, and that this condition can be unrealistic in many problems. To avoid a dependence on the dimension of the problem, their lower bound proof requires making assumptions about the form of iterate updates made by the optimization algorithm. They define the notion of an oblivious, CLI algorithm, where updates are linear functions of previous iterates and oracle results, with parameters that are fixed (or scheduled) before the algorithm begins. In this framework, they show that for a certain class of functions parametrized by \eta, the kth iterate of the algorithm is an order-k polynomial in \eta. From here, they use existing function approximation theory to prove the existence of a function in their class for which the algorithm converges slowly to the optimum, completing the lower bound proof.

Qualitative Assessment

This was a very well-presented paper. It clearly presents a shortcoming of current lower bounds for finite sum optimization (that they apply only in large dimensions) and addresses it with a lower bound which is independent of the dimension of the problem. There are only a few very small issues with the presentation (I will mention those at the end of this section), and overall it is very clear. This paper is an extension of [5] from single function optimization to finite sum, and uses the same analysis techniques in its proofs. Thus, it does not seem to be a massive leap forward in terms of novelty. Nevertheless, the authors' result here on finite sum optimization demonstrates the limitations of oblivious optimization algorithms, and clears up any questions about the complexity of finite sum optimization in low dimensions for many currently hot algorithms. Two small mistakes: - on line 193 in equation (10), C \in R^d rather than C \in R^{dxd} - on line 202 in equation (12), it should be q^T w rather than q

Confidence in this Review

3-Expert (read the paper in detail, know the area, quite certain of my opinion)


Reviewer 6

Summary

The authors use the approximation theory of polynomials to construct a new lower bound for convex optimization problems with finite sum. Further, those lower bounds hold beyond O(n/d) iterations, compared to the state of the art analysis.

Qualitative Assessment

There are 2 major contributions of this work: 1. It provides lower bounds that holds beyond O(n/d) iterations. 2. It extends the current framework in [5] to the coordinate descent method. The proof is long possibly because it borrows some results from the approximation theory, but they are rigorous and self-contained. Thus, I would recommend the paper to be accepted. One possible drawback of the paper with the p-CLI analysis is that it doesn't cover proximal methods. Minor comments: 1. (line 170) Please also cite a textbook with theorem number. The original paper is hard to find on the Internet. 2. Please also give a reference to Yao's principle. 3. Please hint that the Omega(n) in Theorem 2 comes from Thm 5. 4. (Appendix line 317) Should be 1/2 instead of 1/4. 5. (Appendix line 354) I think last 3 line are equalities. 6. (Appendix line 381, last ineq) Why (n(n+2))^2 instead of (n+2)^2? 7. (Appendix line 545) unnecessary comma in the equation 8. (Appendix line 560) Please hint that you use (\alpha+2)e^(-2(\alpha+1)) \geq 1/e^2 9. (Appendix line 570) The first ineq is wrong. But you can still prove it by showing (x-1)(-2x^2+x^2-1) < 0.

Confidence in this Review

2-Confident (read it all; understood it all reasonably well)